# ADAPTIVE FEDERATED OPTIMIZATION

**Sashank J. Reddi**,* **Zachary Charles**,* **Manzil Zaheer, Zachary Garrett, Keith Rush,**
**Jakub Konečný, Sanjiv Kumar, H. Brendan McMahan**
Google Research
`{sashank, zachcharles, manzilzaheer, zachgarrett, krush,`
`konkey, sanjivk, mcmahan}@google.com`

## ABSTRACT

Federated learning is a distributed machine learning paradigm in which a large number of clients coordinate with a central server to learn a model without sharing their own training data. Standard federated optimization methods such as Federated Averaging (FEDAVG) are often difficult to tune and exhibit unfavorable convergence behavior. In non-federated settings, adaptive optimization methods have had notable success in combating such issues. In this work, we propose federated versions of adaptive optimizers, including ADAGRAD, ADAM, and YOGI, and analyze their convergence in the presence of heterogeneous data for general nonconvex settings. Our results highlight the interplay between client heterogeneity and communication efficiency. We also perform extensive experiments on these methods and show that the use of adaptive optimizers can significantly improve the performance of federated learning.

## 1 INTRODUCTION

Federated learning (FL) is a machine learning paradigm in which multiple clients cooperate to learn a model under the orchestration of a central server (McMahan et al., 2017). In FL, raw client data is never shared with the server or other clients. This distinguishes FL from traditional distributed optimization, and requires contending with heterogeneous data. FL has two primary settings, *cross-silo* (eg. FL between large institutions) and *cross-device* (eg. FL across edge devices) (Kairouz et al., 2019, Table 1). In cross-silo FL, most clients participate in every round and can maintain *state* between rounds. In the more challenging cross-device FL, our primary focus, only a small fraction of clients participate in each round, and clients cannot maintain state across rounds. For a more in-depth discussion of FL and the challenges involved, we defer to Kairouz et al. (2019) and Li et al. (2019a).

Standard optimization methods, such as distributed SGD, are often unsuitable in FL and can incur high communication costs. To remedy this, many federated optimization methods use *local client updates*, in which clients update their models multiple times before communicating with the server. This can greatly reduce the amount of communication required to train a model. One such method is FEDAVG (McMahan et al., 2017), in which clients perform multiple epochs of SGD on their local datasets. The clients communicate their models to the server, which averages them to form a new global model. While FEDAVG has seen great success, recent works have highlighted its convergence issues in some settings (Karimireddy et al., 2019; Hsu et al., 2019). This is due to a variety of factors including (1) *client drift* (Karimireddy et al., 2019), where local client models move away from globally optimal models, and (2) a lack of *adaptivity*. FEDAVG is similar in spirit to SGD, and may be unsuitable for settings with heavy-tail stochastic gradient noise distributions, which often arise when training language models (Zhang et al., 2019a). Such settings benefit from adaptive learning rates, which incorporate knowledge of past iterations to perform more informed optimization.

In this paper, we focus on the second issue and present a simple framework for incorporating adaptivity in FL. In particular, we propose a general optimization framework in which (1) clients perform multiple epochs of training using a *client optimizer* to minimize loss on their local data and (2) server updates its global model by applying a gradient-based *server optimizer* to the average of the clients' *model updates*. We show that FEDAVG is the special case where SGD is used as both client and server optimizer and server learning rate is 1. This framework can also seamlessly incorporate

---

*Authors contributed equally to this work

adaptivity by using adaptive optimizers as client or server optimizers. Building upon this, we develop novel adaptive optimization techniques for FL by using per-coordinate methods as server optimizers. By focusing on adaptive server optimization, we enable use of adaptive learning rates without increase in client storage or communication costs, and ensure compatibility with cross-device FL.

**Main contributions**  In light of the above, we highlight the main contributions of the paper.

- We study a general framework for federated optimization using server and client optimizers. This framework generalizes many existing federated optimization methods, including FEDAVG.

- We use this framework to design novel, cross-device compatible, adaptive federated optimization methods, and provide convergence analysis in general nonconvex settings. To the best of our knowledge, these are the first methods for FL using *adaptive server optimization*. We show an important interplay between the number of local steps and the heterogeneity among clients.

- We introduce comprehensive and reproducible empirical benchmarks for comparing federated optimization methods. These benchmarks consist of seven diverse and representative FL tasks involving both image and text data, with varying amounts of heterogeneity and numbers of clients.

- We demonstrate strong empirical performance of our adaptive optimizers throughout, improving upon commonly used baselines. Our results show that our methods can be easier to tune, and highlight their utility in cross-device settings.

**Related work**  FEDAVG was first introduced by McMahan et al. (2017), who showed it can dramatically reduce communication costs. Many variants have since been proposed to tackle issues such as convergence and client drift. Examples include adding a regularization term in the client objectives towards the broadcast model (Li et al., 2018), and server momentum (Hsu et al., 2019). When clients are homogeneous, FEDAVG reduces to local SGD (Zinkevich et al., 2010), which has been analyzed by many works (Stich, 2019; Yu et al., 2019; Wang & Joshi, 2018; Stich & Karimireddy, 2019; Basu et al., 2019). In order to analyze FEDAVG in heterogeneous settings, many works derive convergence rates depending on the amount of heterogeneity (Li et al., 2018; Wang et al., 2019; Khaled et al., 2019; Li et al., 2019b). Typically, the convergence rate of FEDAVG gets worse with client heterogeneity. By using control variates to reduce client drift, the SCAFFOLD method (Karimireddy et al., 2019) achieves convergence rates that are independent of the amount of heterogeneity. While effective in cross-silo FL, the method is incompatible with cross-device FL as it requires clients to maintain state across rounds. For more detailed comparisons, we defer to Kairouz et al. (2019).

Adaptive methods have been the subject of significant theoretical and empirical study, in both convex (McMahan & Streeter, 2010b; Duchi et al., 2011; Kingma & Ba, 2015) and non-convex settings (Li & Orabona, 2018; Ward et al., 2018; Wu et al., 2019). Reddi et al. (2019); Zaheer et al. (2018) study convergence failures of ADAM in certain non-convex settings, and develop an adaptive optimizer, YOGI, designed to improve convergence. While most work on adaptive methods focuses on non-FL settings, Xie et al. (2019) propose ADAALTER, a method for FL using adaptive client optimization. Conceptually, our approach is also related to the LOOKAHEAD optimizer (Zhang et al., 2019b), which was designed for non-FL settings. Similar to ADAALTER, an adaptive FL variant of LOOKAHEAD entails adaptive client optimization (see Appendix B.3 for more details). We note that both ADAALTER and LOOKAHEAD are, in fact, special cases of our framework (see Algorithm 1) and the primary novelty of our work comes in focusing on adaptive *server optimization*. This allows us to avoid aggregating optimizer states across clients, making our methods require at most half as much communication and client memory usage per round (see Appendix B.3 for details).

**Notation**  For $a, b \in \mathbb{R}^d$, we let $\sqrt{a}, a^2$ and $a/b$ denote the element-wise square root, square, and division of the vectors. For $\theta_i \in \mathbb{R}^d$, we use both $\theta_{i,j}$ and $[\theta_i]_j$ to denote its $j^{\text{th}}$ coordinate.

## 2  FEDERATED LEARNING AND FEDAVG

In federated learning, we solve an optimization problem of the form:

$$\min_{x \in \mathbb{R}^d} f(x) = \frac{1}{m} \sum_{i=1}^{m} F_i(x), \tag{1}$$

where $F_i(x) = \mathbb{E}_{z \sim \mathcal{D}_i}[f_i(x, z)]$, is the loss function of the $i^{\text{th}}$ client, $z \in \mathcal{Z}$, and $\mathcal{D}_i$ is the data distribution for the $i^{\text{th}}$ client. For $i \neq j$, $\mathcal{D}_i$ and $\mathcal{D}_j$ may be very different. The functions $F_i$ (and

therefore $f$) may be nonconvex. For each $i$ and $x$, we assume access to an *unbiased* stochastic gradient $g_i(x)$ of the client's true gradient $\nabla F_i(x)$. In addition, we make the following assumptions.

**Assumption 1** (Lipschitz Gradient). *The function $F_i$ is $L$-smooth for all $i \in [m]$ i.e., $\|\nabla F_i(x) - \nabla F_i(y)\| \le L\|x - y\|$, for all $x, y \in \mathbb{R}^d$.*

**Assumption 2** (Bounded Variance). *The function $F_i$ have $\sigma_l$-bounded (local) variance i.e., $\mathbb{E}[\|\nabla[f_i(x, z)]_j - [\nabla F_i(x)]_j\|^2] = \sigma_{l,j}^2$ for all $x \in \mathbb{R}^d$, $j \in [d]$ and $i \in [m]$. Furthermore, we assume the (global) variance is bounded, $(1/m)\sum_{i=1}^m \|\nabla[F_i(x)]_j - [\nabla f(x)]_j\|^2 \le \sigma_{g,j}^2$ for all $x \in \mathbb{R}^d$ and $j \in [d]$.*

**Assumption 3** (Bounded Gradients). *The function $f_i(x, z)$ have $G$-bounded gradients i.e., for any $i \in [m]$, $x \in \mathbb{R}^d$ and $z \in \mathcal{Z}$ we have $|[\nabla f_i(x, z)]_j| \le G$ for all $j \in [d]$.*

With a slight abuse of notation, we use $\sigma_l^2$ and $\sigma_g^2$ to denote $\sum_{j=1}^d \sigma_{l,j}^2$ and $\sum_{j=1}^d \sigma_{g,j}^2$. Assumptions 1 and 3 are fairly standard in nonconvex optimization literature (Reddi et al., 2016; Ward et al., 2018; Zaheer et al., 2018). We make no further assumptions regarding the similarity of clients datasets. Assumption 2 is a form of bounded variance, but between the client objective functions and the overall objective function. This assumption has been used throughout various works on federated optimization (Li et al., 2018; Wang et al., 2019). Intuitively, the parameter $\sigma_g$ quantifies similarity of client objective functions. Note $\sigma_g = 0$ corresponds to the *i.i.d.* setting.

A common approach to solving (1) in federated settings is FEDAVG (McMahan et al., 2017). At each round of FEDAVG, a subset of clients are selected (typically randomly) and the server broadcasts its global model to each client. In parallel, the clients run SGD on their own loss function, and send the resulting model to the server. The server then updates its global model as the average of these local models. See Algorithm 3 in the appendix for more details.

Suppose that at round $t$, the server has model $x_t$ and samples a set $\mathcal{S}$ of clients. Let $x_i^t$ denote the model of each client $i \in \mathcal{S}$ after local training. We rewrite FEDAVG's update as

$$x_{t+1} = \frac{1}{|\mathcal{S}|} \sum_{i \in \mathcal{S}} x_i^t = x_t - \frac{1}{|\mathcal{S}|} \sum_{i \in \mathcal{S}} \left( x_t - x_i^t \right).$$

Let $\Delta_i^t := x_i^t - x_t$ and $\Delta_t := (1/|S|)\sum_{i \in \mathcal{S}} \Delta_i^t$. Then the server update in FEDAVG is equivalent to applying SGD to the "pseudo-gradient" $-\Delta_t$ with learning rate $\eta = 1$. This formulation makes it clear that other choices of $\eta$ are possible. One could also utilize optimizers other than SGD on the clients, or use an alternative update rule on the server. This family of algorithms, which we refer to collectively as FEDOPT, is formalized in Algorithm 1.

---

**Algorithm 1 FEDOPT**

---

1: Input: $x_0$, CLIENTOPT, SERVEROPT
2: **for** $t = 0, \cdots, T - 1$ **do**
3:     Sample a subset $\mathcal{S}$ of clients
4:     $x_{i,0}^t = x_t$
5:     **for** each client $i \in \mathcal{S}$ **in parallel do**
6:         **for** $k = 0, \cdots, K - 1$ **do**
7:             Compute an unbiased estimate $g_{i,k}^t$ of $\nabla F_i(x_{i,k}^t)$
8:             $x_{i,k+1}^t = \text{CLIENTOPT}(x_{i,k}^t, g_{i,k}^t, \eta_l, t)$
9:         $\Delta_i^t = x_{i,K}^t - x_t$
10:     $\Delta_t = \frac{1}{|\mathcal{S}|}\sum_{i \in \mathcal{S}} \Delta_i^t$
11:     $x_{t+1} = \text{SERVEROPT}(x_t, -\Delta_t, \eta, t)$

---

In Algorithm 1, CLIENTOPT and SERVEROPT are *gradient-based* optimizers with learning rates $\eta_l$ and $\eta$ respectively. Intuitively, CLIENTOPT aims to minimize (1) based on each client's local data while SERVEROPT optimizes from a global perspective. FEDOPT naturally allows the use of adaptive optimizers (eg. ADAM, YOGI, etc.), as well as techniques such as server-side momentum (leading to FEDAVGM, proposed by Hsu et al. (2019)). In its most general form, FEDOPT uses a CLIENTOPT whose updates can depend on globally aggregated statistics (e.g. server updates in the

previous iterations). We also allow $\eta$ and $\eta_l$ to depend on the round $t$ in order to encompass learning rate schedules. While we focus on specific adaptive optimizers in this work, we can in principle use any adaptive optimizer (e.g. AMSGRAD (Reddi et al., 2019), ADABOUND (Luo et al., 2019)).

While FEDOPT has intuitive benefits over FEDAVG, it also raises a fundamental question: *Can the negative of the average model difference $\Delta_t$ be used as a pseudo-gradient in general server optimizer updates?* In this paper, we provide an affirmative answer to this question by establishing a theoretical basis for FEDOPT. We will show that the use of the term SERVEROPT is justified, as we can guarantee convergence across a wide variety of server optimizers, including ADAGRAD, ADAM, and YOGI, thus developing principled adaptive optimizers for FL based on our framework.

## 3 ADAPTIVE FEDERATED OPTIMIZATION

In this section, we specialize FEDOPT to settings where SERVEROPT is an adaptive optimization method (one of ADAGRAD, YOGI or ADAM) and CLIENTOPT is SGD. By using adaptive methods (which generally require maintaining state) on the server and SGD on the clients, we ensure our methods have the same communication cost as FEDAVG and work in cross-device settings.

Algorithm 2 provides pseudo-code for our methods. An alternate version using batched data and example-based weighting (as opposed to uniform weighting) of clients is given in Algorithm 5. The parameter $\tau$ controls the algorithms' *degree of adaptivity*, with smaller values of $\tau$ representing higher degrees of adaptivity. Note that the server updates of our methods are invariant to fixed multiplicative changes to the client learning rate $\eta_l$ for appropriately chosen $\tau$, though as we shall see shortly, we will require $\eta_l$ to be sufficiently small in our analysis.

---

**Algorithm 2** FEDADAGRAD , FEDYOGI , and FEDADAM

1: Initialization: $x_0, v_{-1} \geq \tau^2$, decay parameters $\beta_1, \beta_2 \in [0, 1)$
2: **for** $t = 0, \cdots, T - 1$ **do**
3:     Sample subset $\mathcal{S}$ of clients
4:     $x_{i,0}^t = x_t$
5:     **for** each client $i \in \mathcal{S}$ **in parallel do**
6:         **for** $k = 0, \cdots, K - 1$ **do**
7:             Compute an unbiased estimate $g_{i,k}^t$ of $\nabla F_i(x_{i,k}^t)$
8:             $x_{i,k+1}^t = x_{i,k}^t - \eta_l g_{i,k}^t$
9:         $\Delta_i^t = x_{i,K}^t - x_t$
10:     $\Delta_t = \beta_1 \Delta_{t-1} + (1 - \beta_1) \left( \frac{1}{|\mathcal{S}|} \sum_{i \in \mathcal{S}} \Delta_i^t \right)$
11:     $v_t = v_{t-1} + \Delta_t^2$ **(FEDADAGRAD)**
12:     $v_t = v_{t-1} - (1 - \beta_2)\Delta_t^2 \,\mathrm{sign}(v_{t-1} - \Delta_t^2)$ **(FEDYOGI)**
13:     $v_t = \beta_2 v_{t-1} + (1 - \beta_2)\Delta_t^2$ **(FEDADAM)**
14:     $x_{t+1} = x_t + \eta \frac{\Delta_t}{\sqrt{v_t} + \tau}$

---

We provide convergence analyses of these methods in general nonconvex settings, assuming *full participation*, i.e. $\mathcal{S} = [m]$. For expository purposes, we assume $\beta_1 = 0$, though our analysis can be directly extended to $\beta_1 > 0$. Our analysis can also be extended to *partial participation* (i.e. $|\mathcal{S}| < m$, see Appendix A.2.1 for details). Furthermore, non-uniform weighted averaging typically used in FEDAVG (McMahan et al., 2017) can also be incorporated into our analysis fairly easily.

**Theorem 1.** *Let Assumptions 1 to 3 hold, and let $L, G, \sigma_l, \sigma_g$ be as defined therein. Let $\sigma^2 = \sigma_l^2 + 6K\sigma_g^2$. Consider the following conditions for $\eta_l$:*

$$(\text{Condition I}) \qquad \eta_l \leq \frac{1}{K} \min\left\{ \frac{1}{16L}, \frac{1}{T^{1/6}} \left[ \frac{\tau}{120L^2 G} \right]^{1/3} \right\},$$

$$(\text{Condition II}) \qquad \eta_l \leq \frac{1}{3K} \min\left\{ \frac{1}{T^{1/10}} \left[ \frac{\tau^3}{L^2 G^3} \right]^{1/5}, \frac{1}{T^{1/8}} \left[ \frac{\tau^2}{L^3 G \eta} \right]^{1/4} \right\}.$$

*Then the iterates of Algorithm 2 for* FEDADAGRAD *satisfy*

*under Condition I only,*
$$\min_{0 \le t \le T-1} \mathbb{E}\|\nabla f(x_t)\|^2 \le \mathcal{O}\left(\left[\frac{G}{\sqrt{T}} + \frac{\tau}{\eta_l KT}\right](\Psi + \Psi_{var})\right),$$

*under both Condition I & II,*
$$\min_{0 \le t \le T-1} \mathbb{E}\|\nabla f(x_t)\|^2 \le \mathcal{O}\left(\left[\frac{G}{\sqrt{T}} + \frac{\tau}{\eta_l KT}\right]\left(\Psi + \widetilde{\Psi}_{\mathrm{var}}\right)\right).$$

*Here, we define*

$$\Psi = \frac{f(x_0) - f(x^*)}{\eta} + \frac{5\eta_l^3 K^2 L^2 T}{2\tau}\sigma^2,$$

$$\Psi_{\mathrm{var}} = \frac{d(\eta_l KG^2 + \tau\eta L)}{\tau}\left[1 + \log\frac{\tau^2 + \eta_l^2 K^2 G^2 T}{\tau^2}\right],$$

$$\widetilde{\Psi}_{\mathrm{var}} = \frac{2\eta_l KG^2 + \tau\eta L}{\tau^2}\left[\frac{2\eta_l^2 KT}{m}\sigma_l^2 + 10\eta_l^4 K^3 L^2 T\sigma^2\right].$$

All proofs are relegated to Appendix A due to space constraints. When $\eta_l$ satisfies the condition in the second part the above result, we obtain a convergence rate depending on $\min\{\Psi_{\mathrm{var}}, \widetilde{\Psi}_{\mathrm{var}}\}$. To obtain an explicit dependence on $T$ and $K$, we simplify the above result for a specific choice of $\eta, \eta_l$ and $\tau$.

**Corollary 1.** *Suppose $\eta_l$ is such that the conditions in Theorem 1 are satisfied and $\eta_l = \Theta(1/(KL\sqrt{T}))$. Also suppose $\eta = \Theta(\sqrt{Km})$ and $\tau = G/L$. Then, for sufficiently large $T$, the iterates of Algorithm 2 for* FEDADAGRAD *satisfy*

$$\min_{0 \le t \le T-1} \mathbb{E}\|\nabla f(x_t)\|^2 = \mathcal{O}\left(\frac{f(x_0) - f(x^*)}{\sqrt{mKT}} + \frac{2\sigma_l^2 L}{G^2\sqrt{mKT}} + \frac{\sigma^2}{GKT} + \frac{\sigma^2 L\sqrt{m}}{G^2\sqrt{K}T^{3/2}}\right).$$

We defer a detailed discussion about our analysis and its implication to the end of the section.

**Analysis of FEDADAM** Next, we provide the convergence analysis of FEDADAM. The proof of FEDYOGI is very similar and hence, we omit the details of FEDYOGI's analysis.

**Theorem 2.** *Let Assumptions 1 to 3 hold, and $L, G, \sigma_l, \sigma_g$ be as defined therein. Let $\sigma^2 = \sigma_l^2 + 6K\sigma_g^2$. Suppose the client learning rate satisfies $\eta_l \le 1/16LK$ and*

$$\eta_l \le \frac{1}{6K}\min\left\{\left[\frac{\tau}{GL}\right]^{1/2}, \left[\frac{\tau^2}{GL^3\eta}\right]^{1/4}, \left[\frac{\tau}{GL^2}\right]^{1/3}\right\}.$$

*Then the iterates of Algorithm 2 for* FEDADAM *satisfy*

$$\min_{0 \le t \le T-1} \mathbb{E}\|\nabla f(x_t)\|^2 = \mathcal{O}\left(\frac{\sqrt{\beta_2}\eta_l KG + \tau}{\eta_l KT}(\Psi + \Psi_{\mathrm{var}})\right),$$

*where*

$$\Psi = \frac{f(x_0) - f(x^*)}{\eta} + \frac{5\eta_l^3 K^2 L^2 T}{2\tau}\sigma^2,$$

$$\Psi_{var} = \left(G + \frac{\eta L}{2}\right)\left[\frac{4\eta_l^2 KT}{m\tau^2}\sigma_l^2 + \frac{20\eta_l^4 K^3 L^2 T}{\tau^2}\sigma^2\right].$$

Similar to the FEDADAGRAD case, we restate the above result for a specific choice of $\eta_l, \eta$ and $\tau$ in order to highlight the dependence of $K$ and $T$.

**Corollary 2.** *Suppose $\eta_l$ is chosen such that the conditions in Theorem 2 are satisfied and that $\eta_l = \Theta(1/(KL\sqrt{T}))$. Also, suppose $\eta = \Theta(\sqrt{Km})$ and $\tau = G/L$. Then, for sufficiently large $T$, the iterates of Algorithm 2 for* FEDADAM *satisfy*

$$\min_{0 \le t \le T-1} \mathbb{E}\|\nabla f(x_t)\|^2 = \mathcal{O}\left(\frac{f(x_0) - f(x^*)}{\sqrt{mKT}} + \frac{2\sigma_l^2 L}{G^2\sqrt{mKT}} + \frac{\sigma^2}{GKT} + \frac{\sigma^2 L\sqrt{m}}{G^2\sqrt{K}T^{3/2}}\right).$$

**Remark 1.** *The server learning rate $\eta = 1$ typically used in* FEDAVG *does not necessarily minimize the upper bound in Theorems 1 & 2. The effect of $\sigma_g$, a measure of client heterogeneity, on convergence can be reduced by choosing sufficiently $\eta_l$ and a reasonably large $\eta$ (e.g. see Corollary 1). Thus, the effect of client heterogeneity can be reduced by carefully choosing client and server learning rates, but not removed entirely. Our empirical analysis (eg. Figure 1) supports this conclusion.*

**Discussion.** We briefly discuss our theoretical analysis and its implications in the FL setting. The convergence rates for FEDADAGRAD and FEDADAM are similar, so our discussion applies to all the adaptive federated optimization algorithms (including FEDYOGI) proposed in the paper.

(i) **Comparison of convergence rates.** When $T$ is sufficiently large compared to $K$, $\mathcal{O}(1/\sqrt{mKT})$ is the dominant term in Corollary 1 & 2. Thus, we effectively obtain a convergence rate of $\mathcal{O}(1/\sqrt{mKT})$, which matches the *best known rate* for the general non-convex setting of our interest (e.g. see (Karimireddy et al., 2019)). We also note that in the $i.i.d$ setting considered in (Wang & Joshi, 2018), which corresponds to $\sigma_g = 0$, we match their convergence rates. Similar to the centralized setting, it is possible to obtain convergence rates with better dependence on constants for federated adaptive methods, compared to FEDAVG, by incorporating non-uniform bounds on gradients across coordinates (Zaheer et al., 2018).

(ii) **Learning rates & their decay.** The client learning rate of $1/\sqrt{T}$ in our analysis requires knowledge of the number of rounds $T$ a priori; however, it is easy to generalize our analysis to the case where $\eta_l$ is decayed at a rate of $1/\sqrt{t}$. Observe that one must decay $\eta_l$, not the server learning rate $\eta$, to obtain convergence. This is because the *client drift* introduced by the local updates does not vanish as $T \to \infty$ when $\eta_l$ is constant. As we show in Appendix E.6, learning rate decay can improve empirical performance. Also, note the inverse relationship between $\eta_l$ and $\eta$ in Corollary 1 & 2, which we observe in our empirical analysis (see Appendix E.4).

(iii) **Communication efficiency & local steps.** The total communication cost of the algorithms depends on the number of communication rounds $T$. From Corollary 1 & 2, it is clear that a larger $K$ leads to fewer rounds of communication as long as $K = \mathcal{O}(T\sigma_l^2/\sigma_g^2)$. Thus, the number of local iterations can be large when either the ratio $\sigma_l^2/\sigma_g^2$ or $T$ is large. In the $i.i.d$ setting where $\sigma_g = 0$, unsurprisingly, $K$ can be very large.

(iv) **Client heterogeneity.** While careful selection of client and server learning rates can reduce the effect of client heterogeneity (see Remark 1), it *does not* completely remove it. In highly heterogeneous settings, it may be necessary to use mechanisms such as control variates (Karimireddy et al., 2019). However, our empirical analysis suggest that for moderate, naturally arising heterogeneity, adaptive optimizers are quite effective, especially in cross-device settings (see Figure 1). Furthermore, our algorithms can be directly combined with such mechanisms.

As mentioned earlier, for the sake of simplicity, our analysis assumes full-participation ($\mathcal{S} = [m]$). Our analysis can be directly generalized to limited participation at the cost of an additional variance term in our rates that depends on $|\mathcal{S}|/m$, the fraction of clients sampled (see Section A.2.1 for details).

## 4 EXPERIMENTAL EVALUATION: DATASETS, TASKS, AND METHODS

We evaluate our algorithms on what we believe is the most extensive and representative suite of federated datasets and modeling tasks to date. We wish to understand how server adaptivity can help improve convergence, especially in cross-device settings. To accomplish this, we conduct simulations on seven diverse and representative learning tasks across five datasets. Notably, three of the five have a naturally-arising client partitioning, highly representative of real-world FL problems.

**Datasets, models, and tasks** We use five datasets: CIFAR-10, CIFAR-100 (Krizhevsky & Hinton, 2009), EMNIST (Cohen et al., 2017), Shakespeare (McMahan et al., 2017), and Stack Overflow (Authors, 2019). The first three are image datasets, the last two are text datasets. For CIFAR-10 and CIFAR-100, we train ResNet-18 (replacing batch norm with group norm (Hsieh et al., 2019)). For EMNIST, we train a CNN for character recognition (EMNIST CR) and a bottleneck autoencoder (EMNIST AE). For Shakespeare, we train an RNN for next-character-prediction. For Stack Overflow, we perform tag prediction using logistic regression on bag-of-words vectors (SO LR) and train an RNN to do next-word-prediction (SO NWP). For full details of the datasets, see Appendix C.

**Implementation** We implement all algorithms in TensorFlow Federated (Ingerman & Ostrowski, 2019). Clients are sampled uniformly at random, without replacement in a given round, but with replacement across rounds. Our implementation has two important characteristics. First, instead of

doing $K$ training steps per client, we do $E$ *epochs* of training over each client's dataset. Second, to account for varying numbers of gradient steps per client, we weight the average of the client outputs $\Delta_i^t$ by each client's number of training samples. This follows the approach of (McMahan et al., 2017), and can often outperform uniform weighting (Zaheer et al., 2018). For full descriptions of the algorithms used, see Appendix B.

**Optimizers and hyperparameters** We compare FEDADAGRAD, FEDADAM, and FEDYOGI (with adaptivity $\tau$) to FEDOPT where CLIENTOPT and SERVEROPT are SGD with learning rates $\eta_l$ and $\eta$. For the server, we use a momentum parameter of 0 (FEDAVG), and 0.9 (FEDAVGM). We fix the client batch size on a per-task level (see Appendix D.3). For FEDADAM and FEDYOGI, we fix a momentum parameter $\beta_1 = 0.9$ and a second moment parameter $\beta_2 = 0.99$. We also compare to SCAFFOLD (see Appendix B.2 for implementation details). For SO NWP, we sample 50 clients per round, while for all other tasks we sample 10. We use $E = 1$ local epochs throughout.

We select $\eta_l$, $\eta$, and $\tau$ by grid-search tuning. While this is often done using validation data in centralized settings, such data is often inaccessible in FL, especially cross-device FL. Therefore, we tune by selecting the parameters that minimize *the average training loss over the last 100 rounds of training*. We run 1500 rounds of training on the EMNIST CR, Shakespeare, and Stack Overflow tasks, 3000 rounds for EMNIST AE, and 4000 rounds for the CIFAR tasks. For more details and a record of the best hyperparameters, see Appendix D.

**Validation metrics** For all tasks, we measure the performance on a validation set throughout training. For Stack Overflow tasks, the validation set contains 10,000 randomly sampled test examples (due to the size of the test dataset, see Table 2). For all other tasks, we use the entire test set. Since all algorithms exchange equal-sized objects between server and clients, we use the number of communication rounds as a proxy for wall-clock training time.

## 5 EXPERIMENTAL EVALUATION: RESULTS

### 5.1 COMPARISONS BETWEEN METHODS

We compare the convergence of our adaptive methods to non-adaptive methods: FEDAVG, FEDAVGM and SCAFFOLD. Plots of validation performances for each task/optimizer are in Figure 1, and Table 1 summarizes the last-100-round validation performance. Due to space constraints, results for EMNIST CR are in Appendix E.1, and full test set results for Stack Overflow are in Appendix E.2.

**Sparse-gradient tasks** Text data often produces long-tailed feature distributions, leading to approximately-sparse gradients which adaptive optimizers can capitalize on (Zhang et al., 2019a). Both Stack Overflow tasks exhibit such behavior, though they are otherwise dramatically different—in feature representation (bag-of-words vs. variable-length token sequence), model architecture (GLM vs deep network), and optimization landscape (convex vs nonconvex). In both tasks, words that do not appear in a client's dataset produce nearzero client updates. Thus, the accumulators $v_{t,j}$ in Algorithm 2 remain small for parameters tied to rare words, allowing large updates to be made when they do occur.

| FED... | ADAGRAD | ADAM | YOGI | AVGM | AVG |
|---|---|---|---|---|---|
| CIFAR-10 | 72.1 | 77.4 | **78.0** | 77.4 | 72.8 |
| CIFAR-100 | 47.9 | **52.5** | 52.4 | 52.4 | 44.7 |
| EMNIST CR | 85.1 | **85.6** | 85.5 | 85.2 | 84.9 |
| SHAKESPEARE | **57.5** | 57.0 | 57.2 | 57.3 | 56.9 |
| SO NWP | 23.8 | **25.2** | 25.2 | 23.8 | 19.5 |
| SO LR | **67.1** | 65.8 | 65.9 | 36.9 | 30.0 |
| EMNIST AE | 4.20 | 1.01 | **0.98** | 1.65 | 6.47 |

Table 1: Average validation performance over the last 100 rounds: % accuracy for rows 1–5; Recall@5 ($\times 100$) for Stack Overflow LR; and MSE ($\times 1000$) for EMNIST AE. Performance within $0.5\%$ of the best result for each task are shown in bold.

This intuition is born out in Figure 1, where adaptive optimizers dramatically outperform non-adaptive ones. For the non-convex NWP task, momentum is also critical, whereas it slightly hinders performance for the convex LR task.

**Dense-gradient tasks** CIFAR-10/100, EMNIST AE/CR, and Shakespeare lack a sparse-gradient structure. Shakespeare is relatively easy—most optimizers perform well after enough rounds *once suitably tuned*, though FEDADAGRAD converges faster. For CIFAR-10/100 and EMNIST AE, adaptivity and momentum offer substantial improvements over FEDAVG. Moreover, FEDYOGI and FEDADAM have faster initial convergence than FEDAVGM on these tasks. Notably, FEDADAM and FEDYOGI perform comparably to or better than non-adaptive optimizers throughout, and close to or better than

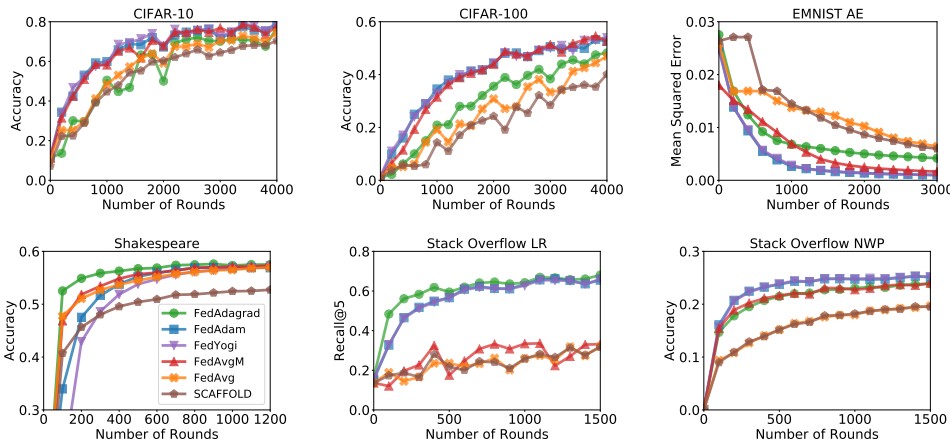

Figure 1: Validation accuracy of adaptive and non-adaptive methods, as well as SCAFFOLD, using constant learning rates $\eta$ and $\eta_l$ tuned to achieve the best training performance over the last 100 communication rounds; see Appendix D.2 for grids.

FEDADAGRAD throughout. As we discuss below, FEDADAM and FEDYOGI actually enable easier learning rate tuning than FEDAVGM in many tasks.

**Comparison to SCAFFOLD** On all tasks, SCAFFOLD performs comparably to or worse than FEDAVG and our adaptive methods. On Stack Overflow, SCAFFOLD and FEDAVG are nearly identical. This is because the number of clients (342,477) makes it unlikely we sample any client more than once. Intuitively, SCAFFOLD does not have a chance to use its client control variates. In other tasks, SCAFFOLD performs worse than other methods. We present two possible explanations: First, we only sample a small fraction of clients at each round, so most users are sampled infrequently. Intuitively, the client control variates can become stale, and may consequently degrade the performance. Second, SCAFFOLD is similar to variance reduction methods such as SVRG (Johnson & Zhang, 2013). While theoretically performant, such methods often perform worse than SGD in practice (Defazio & Bottou, 2018). As shown by Defazio et al. (2014), variance reduction often only accelerates convergence when close to a critical point. In cross-device settings (where the number of communication rounds are limited), SCAFFOLD may actually reduce empirical performance.

## 5.2 EASE OF TUNING

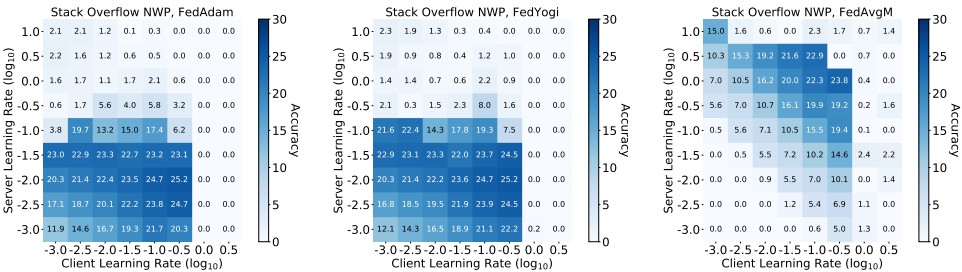

Figure 2: Validation accuracy (averaged over the last 100 rounds) of FEDADAM, FEDYOGI, and FEDAVGM for various client/server learning rates combination on the SO NWP task. For FEDADAM and FEDYOGI, we set $\tau = 10^{-3}$.

Obtaining optimal performance involves tuning $\eta_l, \eta$, and for the adaptive methods, $\tau$. To quantify how easy it is to tune various methods, we plot their validation performance as a function of $\eta_l$ and $\eta$. Figure 2 gives results for FEDADAM, FEDYOGI, and FEDAVGM on Stack Overflow NWP. Plots for all other optimizers and tasks are in Appendix E.3. For FEDAVGM, there are only a few good values of $\eta_l$ for each $\eta$, while for FEDADAM and FEDYOGI, there are many good values of $\eta_l$ for a range of

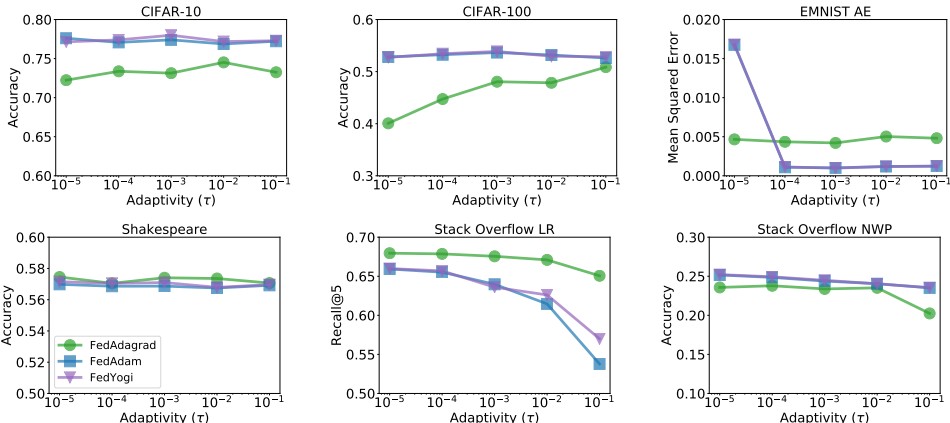

Figure 3: Validation performance of FEDADAGRAD, FEDADAM, and FEDYOGI for varying $\tau$ on various tasks. The learning rates $\eta$ and $\eta_l$ are tuned for each $\tau$ to achieve the best training performance on the last 100 communication rounds.

$\eta$. Thus, FEDADAM and FEDYOGI are arguably easier to tune in this setting. Similar results hold for other tasks and optimizers (Figures 5 to 11).

This leads to a natural question: Is the reduction in the need to tune $\eta_l$ and $\eta$ offset by the need to tune the adaptivity $\tau$? In fact, while we tune $\tau$ in Figure 1, our results are relatively robust to $\tau$. To demonstrate, we plot the best validation performance for various $\tau$ in Figure 3. For nearly all tasks and optimizers, $\tau = 10^{-3}$ works almost as well all other values. This aligns with work by Zaheer et al. (2018), who show that moderately large $\tau$ yield better performance for centralized adaptive optimizers. FEDADAM and FEDYOGI see only small differences in performance among $\tau$ on all tasks except Stack Overflow LR (for which FEDADAGRAD is the best optimizer, and is robust to $\tau$).

## 5.3 OTHER FINDINGS

We present additional empirical analyses in Appendix E. These include EMNIST CR results (Appendix E.1), Stack Overflow results on the full test dataset (Appendix E.2), client/server learning rate heat maps for all optimizers and tasks (Appendix E.3), an analysis of the relationship between $\eta$ and $\eta_l$ (Appendix E.4), and experiments with learning rate decay (Appendix E.6).

## 6 CONCLUSION

In this paper, we demonstrated that adaptive optimizers can be powerful tools in improving the convergence of FL. By using a simple client/server optimizer framework, we can incorporate adaptivity into FL in a principled, intuitive, and theoretically-justified manner. We also developed comprehensive benchmarks for comparing federated optimization algorithms. To encourage reproducibility and breadth of comparison, we have attempted to describe our experiments as rigorously as possible, and have created an open-source framework with all models, datasets, and code. We believe our work raises many important questions about how best to perform federated optimization. Example directions for future research include understanding how the use of adaptivity affects differential privacy and fairness.

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

## A PROOF OF RESULTS

### A.1 MAIN CHALLENGES

We first recap some of the central challenges to our analysis. Theoretical analyses of optimization methods for federated learning are much different than analyses for centralized settings. The key factors complicating the analysis are:

1. Clients performing multiple local updates.

2. Data heterogeneity.

3. Understanding the communication complexity.

As a result of (1), the updates from the clients to the server are not gradients, or even unbiased estimates of gradients, they are *pseudo-gradients* (see Section 2). These pseudo-gradients are challenging to analyze as they can have both high bias (their expectation is not the gradient of the empirical loss function) and high variance (due to compounding variance across client updates) and are therefore challenging to bound. This is exacerbated by (2), which we quantify by the parameter $\sigma_g$ in Section 2. Things are further complicated by (3), as we must obtain a good trade-off between the number of client updates taken per round ($K$ in Algorithms 1 and 2) and the number of communication rounds $T$. Such trade-offs do not exist in centralized optimization.

### A.2 PROOF OF THEOREM 1

*Proof of Theorem 1.* Recall that the server update of FEDADAGRAD is the following

$$x_{t+1,i} = x_{t,i} + \eta \frac{\Delta_{t,i}}{\sqrt{v_{t,i}} + \tau},$$

for all $i \in [d]$. Since the function $f$ is $L$-smooth, we have the following:

$$f(x_{t+1}) \leq f(x_t) + \langle \nabla f(x_t), x_{t+1} - x_t \rangle + \frac{L}{2}\|x_{t+1} - x_t\|^2$$

$$= f(x_t) + \eta \left\langle \nabla f(x_t), \frac{\Delta_t}{\sqrt{v_t} + \tau} \right\rangle + \frac{\eta^2 L}{2} \sum_{i=1}^{d} \frac{\Delta_{t,i}^2}{(\sqrt{v_{t,i}} + \tau)^2} \tag{2}$$

The second step follows simply from FEDADAGRAD's update. We take the expectation of $f(x_{t+1})$ (over randomness at time step $t$) in the above inequality:

$$\mathbb{E}_t[f(x_{t+1})] \leq f(x_t) + \eta \left\langle \nabla f(x_t), \mathbb{E}_t\left[\frac{\Delta_t}{\sqrt{v_t} + \tau}\right]\right\rangle + \frac{\eta^2 L}{2} \sum_{i=1}^{d} \mathbb{E}_t\left[\frac{\Delta_{t,i}^2}{(\sqrt{v_{t,i}} + \tau)^2}\right]$$

$$= f(x_t) + \eta \left\langle \nabla f(x_t), \mathbb{E}_t\left[\frac{\Delta_t}{\sqrt{v_t} + \tau} - \frac{\Delta_t}{\sqrt{v_{t-1}} + \tau} + \frac{\Delta_t}{\sqrt{v_{t-1}} + \tau}\right]\right\rangle$$

$$+ \frac{\eta^2 L}{2} \sum_{j=1}^{d} \mathbb{E}_t\left[\frac{\Delta_{t,j}^2}{(\sqrt{v_{t,j}} + \tau)^2}\right]$$

$$= f(x_t) + \eta \underbrace{\left\langle \nabla f(x_t), \mathbb{E}_t\left[\frac{\Delta_t}{\sqrt{v_{t-1}} + \tau}\right]\right\rangle}_{T_1} + \eta \underbrace{\left\langle \nabla f(x_t), \mathbb{E}_t\left[\frac{\Delta_t}{\sqrt{v_t} + \tau} - \frac{\Delta_t}{\sqrt{v_{t-1}} + \tau}\right]\right\rangle}_{T_2}$$

$$+ \frac{\eta^2 L}{2} \sum_{j=1}^{d} \mathbb{E}_t\left[\frac{\Delta_{t,j}^2}{(\sqrt{v_{t,j}} + \tau)^2}\right] \tag{3}$$

We will first bound $T_2$ in the following manner:

$$T_2 = \left\langle \nabla f(x_t), \mathbb{E}_t \left[ \frac{\Delta_t}{\sqrt{v_t} + \tau} - \frac{\Delta_t}{\sqrt{v_{t-1}} + \tau} \right] \right\rangle$$

$$= \mathbb{E}_t \sum_{j=1}^d [\nabla f(x_t)]_j \times \left[ \frac{\Delta_{t,j}}{\sqrt{v_{t,j}} + \tau} - \frac{\Delta_{t,j}}{\sqrt{v_{t-1,j}} + \tau} \right]$$

$$= \mathbb{E}_t \sum_{j=1}^d [\nabla f(x_t)]_j \times \Delta_{t,j} \times \left[ \frac{\sqrt{v_{t-1,j}} - \sqrt{v_{t,j}}}{(\sqrt{v_{t,j}} + \tau)(\sqrt{v_{t-1,j}} + \tau)}, \right]$$

and recalling $v_t = v_{t-1} + \Delta_t^2$ so $-\Delta_{t,j}^2 = (\sqrt{v_{t-1,j}} - \sqrt{v_{t,j}})(\sqrt{v_{t-1,j}} + \sqrt{v_{t,j}}))$ we have,

$$= \mathbb{E}_t \sum_{j=1}^d [\nabla f(x_t)]_j \times \Delta_{t,j} \times \left[ \frac{-\Delta_{t,j}^2}{(\sqrt{v_{t,j}} + \tau)(\sqrt{v_{t-1,j}} + \tau)(\sqrt{v_{t-1,j}} + \sqrt{v_{t,j}})} \right]$$

$$\leq \mathbb{E}_t \sum_{j=1}^d |\nabla f(x_t)]_j| \times |\Delta_{t,j}| \times \left[ \frac{\Delta_{t,j}^2}{(\sqrt{v_{t,j}} + \tau)(\sqrt{v_{t-1,j}} + \tau)(\sqrt{v_{t-1,j}} + \sqrt{v_{t,j}})} \right]$$

$$\leq \mathbb{E}_t \sum_{j=1}^d |\nabla f(x_t)]_j| \times |\Delta_{t,j}| \times \left[ \frac{\Delta_{t,j}^2}{(v_{t,j} + \tau^2)(\sqrt{v_{t-1,j}} + \tau)} \right] \qquad \text{since } v_{t-1,j} \geq \tau^2.$$

Here $v_{t-1,j} \geq \tau$ since $v_{-1} \geq \tau$ (see the initialization of Algorithm 2) and $v_{t,j}$ is increasing in $t$. The above bound can be further upper bounded in the following manner:

$$T_2 \leq \mathbb{E}_t \sum_{j=1}^d \eta_l K G^2 \left[ \frac{\Delta_{t,j}^2}{(v_{t,j} + \tau^2)(\sqrt{v_{t-1,j}} + \tau)} \right] \qquad \text{since } [\nabla f(x_t)]_i \leq G \text{ and } \Delta_{t,i} \leq \eta_l K G$$

$$\leq \mathbb{E}_t \sum_{j=1}^d \frac{\eta_l K G^2}{\tau} \left[ \frac{\Delta_{t,j}^2}{\sum_{l=0}^t \Delta_{l,j}^2 + \tau^2} \right] \qquad \text{since } \sqrt{v_{t-1,j}} \geq 0. \qquad (4)$$

**Bounding $T_1$** We now turn our attention to bounding the term $T_1$, which we need to be sufficiently negative. We observe the following:

$$T_1 = \left\langle \nabla f(x_t), \mathbb{E}_t \left[ \frac{\Delta_t}{\sqrt{v_{t-1}} + \tau} \right] \right\rangle$$

$$= \left\langle \frac{\nabla f(x_t)}{\sqrt{v_{t-1}} + \tau}, \mathbb{E}_t \left[ \Delta_t - \eta_l K \nabla f(x_t) + \eta_l K \nabla f(x_t) \right] \right\rangle$$

$$= -\eta_l K \sum_{j=1}^d \frac{[\nabla f(x_t)]_j^2}{\sqrt{v_{t-1,j}} + \tau} + \underbrace{\left\langle \frac{\nabla f(x_t)}{\sqrt{v_{t-1}} + \tau}, \mathbb{E}_t \left[ \Delta_t + \eta_l K \nabla f(x_t) \right] \right\rangle}_{T_3}. \qquad (5)$$

In order to bound $T_1$, we use the following upper bound on $T_3$ (which captures the difference between the actual update $\Delta_t$ and an appropriate scaling of $-\nabla f(x_t)$):

$$T_3 = \left\langle \frac{\nabla f(x_t)}{\sqrt{v_{t-1}} + \tau}, \mathbb{E}_t \left[ \Delta_t + \eta_l K \nabla f(x_t) \right] \right\rangle$$

$$= \left\langle \frac{\nabla f(x_t)}{\sqrt{v_{t-1}} + \tau}, \mathbb{E}_t \left[ -\frac{1}{m} \sum_{i=1}^m \sum_{k=0}^{K-1} \eta_l g_{i,k}^t + \eta_l K \nabla f(x_t) \right] \right\rangle$$

$$= \left\langle \frac{\nabla f(x_t)}{\sqrt{v_{t-1}} + \tau}, \mathbb{E}_t \left[ -\frac{1}{m} \sum_{i=1}^m \sum_{k=0}^{K-1} \eta_l \nabla F_i(x_{i,k}^t) + \eta_l K \nabla f(x_t) \right] \right\rangle.$$

Here we used the fact that $\nabla f(x_t) = \frac{1}{m} \sum_{i=1}^{m} \nabla F_i(x_t)$ and $g_{i,k}^t$ is an unbiased estimator of the gradient at $x_{i,k}^t$, we further bound $T_3$ as follows using a simple application of the fact that $ab \leq (a^2 + b^2)/2.$ :

$$
\begin{aligned}
T_3 &\leq \frac{\eta_l K}{2} \sum_{j=1}^{d} \frac{[\nabla f(x_t)]_j^2}{\sqrt{v_{t-1,j}} + \tau} + \frac{\eta_l}{2K} \mathbb{E}_t \left[ \left\| \frac{1}{m} \sum_{i=1}^{m} \sum_{k=0}^{K-1} \frac{\nabla F_i(x_{i,k}^t)}{\sqrt{\sqrt{v_{t-1}} + \tau}} - \frac{1}{m} \sum_{i=1}^{m} \sum_{k=0}^{K-1} \frac{\nabla F_i(x_t)}{\sqrt{\sqrt{v_{t-1}} + \tau}} \right\|^2 \right] \\
&\leq \frac{\eta_l K}{2} \sum_{j=1}^{d} \frac{[\nabla f(x_t)]_j^2}{\sqrt{v_{t-1,j}} + \tau} + \frac{\eta_l}{2m} \mathbb{E}_t \left[ \sum_{i=1}^{m} \sum_{k=0}^{K-1} \left\| \frac{\nabla F_i(x_{i,k}^t) - \nabla F_i(x_t)}{\sqrt{\sqrt{v_{t-1}} + \tau}} \right\|^2 \right] \\
&\leq \frac{\eta_l K}{2} \sum_{j=1}^{d} \frac{[\nabla f(x_t)]_j^2}{\sqrt{v_{t-1,j}} + \tau} + \frac{\eta_l L^2}{2m\tau} \mathbb{E}_t \left[ \sum_{i=1}^{m} \sum_{k=0}^{K-1} \| x_{i,k}^t - x_t \|^2 \right] \quad \text{using Assumption 1 and } v_{t-1} \geq 0.
\end{aligned}
$$
(6)

The second inequality follows from Lemma 6. The last inequality follows from $L$-Lipschitz nature of the gradient (Assumption 1). We now prove a lemma that bounds the "drift" of the $x_{i,k}^t$ from $x_t$:

**Lemma 3.** *For any step-size satisfying $\eta_l \leq \frac{1}{8LK}$, we can bound the drift for any $k \in \{0, \cdots, K-1\}$ as*

$$
\frac{1}{m} \sum_{i=1}^{m} \mathbb{E}\| x_{i,k}^t - x_t \|^2 \leq 5K\eta_l^2 \mathbb{E} \sum_{j=1}^{d} (\sigma_{l,j}^2 + 2K\sigma_{g,j}^2) + 30K^2\eta_l^2 \mathbb{E}[\|\nabla f(x_t)))\|^2].
$$
(7)

*Proof.* The result trivially holds for $k = 1$ since $x_{i,0}^t = x_t$ for all $i \in [m]$. We now turn our attention to the case where $k \geq 1$. To prove the above result, we observe that for any client $i \in [m]$ and $k \in [K]$,

$$
\mathbb{E}\| x_{i,k}^t - x_t \|^2 = \mathbb{E}\| x_{i,k-1}^t - x_t - \eta_l g_{i,k-1}^t \|^2
$$
$$
\leq \mathbb{E}\| x_{i,k-1}^t - x_t - \eta_l(g_{i,k-1}^t - \nabla F_i(x_{i,k-1}^t) + \nabla F_i(x_{i,k-1}^t) - \nabla F_i(x_t) + \nabla F_i(x_t) - \nabla f(x_t) + \nabla f(x_t)) \|^2
$$
$$
\leq \left( 1 + \frac{1}{2K-1} \right) \mathbb{E}\| x_{i,k-1}^t - x_t \|^2 + \mathbb{E}\| \eta_l(g_{i,k-1}^t - \nabla F_i(x_{i,k-1}^t)) \|^2
$$
$$
+ 6K\mathbb{E}[\|\eta_l(\nabla F_i(x_{i,k-1}^t) - \nabla F_i(x_t))\|^2] + 6K\mathbb{E}[\|\eta_l(\nabla F_i(x_t) - \nabla f(x_t))\|^2] + 6K\mathbb{E}[\|\eta_l\nabla f(x_t)))\|^2]
$$

The first inequality uses the fact that $g_{k-1,i}^t$ is an unbiased estimator of $\nabla F_i(x_{i,k-1}^t)$ and Lemma 7. The above quantity can be further bounded by the following:

$$
\mathbb{E}\| x_{i,k}^t - x_t \|^2 \leq \left( 1 + \frac{1}{2K-1} \right) \mathbb{E}\| x_{i,k-1}^t - x_t \|^2 + \eta_l^2 \mathbb{E} \sum_{j=1}^{d} \sigma_{l,j}^2 + 6K\eta_l^2 \mathbb{E}\| L(x_{i,k-1}^t - x_t) \|^2
$$
$$
+ 6K\mathbb{E}[\|\eta_l(\nabla F_i(x_t) - \nabla f(x_t))\|^2] + 6K\mathbb{E}[\|\eta_l\nabla f(x_t)))\|^2]
$$
$$
= \left( 1 + \frac{1}{2K-1} + 6K\eta_l^2 L^2 \right) \mathbb{E}\| (x_{i,k-1}^t - x_t) \|^2 + \eta_l^2 \mathbb{E} \sum_{j=1}^{d} \sigma_{l,j}^2
$$
$$
+ 6K\mathbb{E}[\|\eta_l(\nabla F_i(x_t) - \nabla f(x_t))\|^2] + 6K\eta_l^2 \mathbb{E}[\|\nabla f(x_t)))\|^2]
$$

Here, the first inequality follows from Assumption 1 and 2. Averaging over the clients $i$, we obtain the following:

$$
\frac{1}{m} \sum_{i=1}^{m} \mathbb{E}\| x_{i,k}^t - x_t \|^2 \leq \left( 1 + \frac{1}{2K-1} + 6K\eta_l^2 L^2 \right) \frac{1}{m} \sum_{i=1}^{m} \mathbb{E}\| x_{i,k-1}^t - x_t \|^2 + \eta_l^2 \mathbb{E} \sum_{j=1}^{d} \sigma_{l,j}^2
$$
$$
+ \frac{6K}{m} \sum_{i=1}^{m} \mathbb{E}[\|\eta_l(\nabla F_i(x_t) - \nabla f(x_t))\|^2] + 6K\eta_l^2 \mathbb{E}[\|\nabla f(x_t)))\|^2]
$$
$$
\leq \left( 1 + \frac{1}{2K-1} + 6K\eta_l^2 L^2 \right) \frac{1}{m} \sum_{i=1}^{m} \mathbb{E}\| x_{i,k-1}^t - x_t \|^2 + \eta_l^2 \mathbb{E} \sum_{j=1}^{d} (\sigma_{l,j}^2 + 6K\sigma_{g,j}^2)
$$
$$
+ 6K\eta_l^2 \mathbb{E}[\|\nabla f(x_t)))\|^2]
$$

From the above, we get the following inequality:

$$\frac{1}{m}\sum_{i=1}^{m}\mathbb{E}\|x_{i,k}^{t}-x_{t}\|^{2} \leq \left(1+\frac{1}{K-1}\right)\frac{1}{m}\sum_{i=1}^{m}\mathbb{E}\|x_{i,k-1}^{t}-x_{t}\|^{2}+\eta_{l}^{2}\mathbb{E}\sum_{j=1}^{d}(\sigma_{l,j}^{2}+6K\sigma_{g,j}^{2})$$
$$+6K\eta_{l}^{2}\mathbb{E}[\|\nabla f(x_{t})))\|^{2}]$$

Unrolling the recursion, we obtain the following:

$$\frac{1}{m}\sum_{i=1}^{m}\mathbb{E}\|x_{i,k}^{t}-x_{t}\|^{2} \leq \sum_{p=0}^{k-1}\left(1+\frac{1}{K-1}\right)^{p}\left[\eta_{l}^{2}\mathbb{E}\sum_{j=1}^{d}(\sigma_{l,j}^{2}+6K\sigma_{g,j}^{2})+6K\eta_{l}^{2}\mathbb{E}[\|\nabla f(x_{t})))\|^{2}]\right]$$

$$\leq (K-1)\times\left[\left(1+\frac{1}{K-1}\right)^{K}-1\right]\times\left[\eta_{l}^{2}\mathbb{E}\sum_{j=1}^{d}(\sigma_{l,j}^{2}+6K\sigma_{g,j}^{2})+6K\eta_{l}^{2}\mathbb{E}[\|\nabla f(x_{t})))\|^{2}]\right]$$

$$\leq \left[5K\eta_{l}^{2}\mathbb{E}\sum_{j=1}^{d}(\sigma_{l,j}^{2}+6K\sigma_{g,j}^{2})+30K^{2}\eta_{l}^{2}\mathbb{E}[\|\nabla f(x_{t})))\|^{2}]\right],$$

concluding the proof of Lemma 3. The last inequality uses the fact that $(1+\frac{1}{K-1})^{K}\leq 5$ for $K>1$. $\qquad\square$

Using the above lemma in Equation 6 and Condition I, we can bound $T_3$ in the following manner:

$$T_{3} \leq \frac{\eta_{l}K}{2}\sum_{j=1}^{d}\frac{[\nabla f(x_{t})]_{j}^{2}}{\sqrt{v_{t-1,j}}+\tau}+\frac{\eta_{l}L^{2}}{2m\tau}\mathbb{E}_{t}\left[\sum_{i=1}^{m}\sum_{k=0}^{K-1}\sum_{j=1}^{d}([x_{i,k}^{t}]_{j}-[x_{t}]_{j})^{2}\right]$$

$$\leq \frac{\eta_{l}K}{2}\sum_{j=1}^{d}\frac{[\nabla f(x_{t})]_{j}^{2}}{\sqrt{v_{t-1,j}}+\tau}+\frac{\eta_{l}KL^{2}}{2\tau}\left[5K\eta_{l}^{2}\mathbb{E}\sum_{j=1}^{d}(\sigma_{l,j}^{2}+6K\sigma_{g,j}^{2})+30K^{2}\eta_{l}^{2}\mathbb{E}[\|\nabla f(x_{t})))\|^{2}]\right]$$

$$\leq \frac{3\eta_{l}K}{4}\sum_{j=1}^{d}\frac{[\nabla f(x_{t})]_{j}^{2}}{\sqrt{v_{t-1,j}}+\tau}+\frac{5\eta_{l}^{3}K^{2}L^{2}}{2\tau}\mathbb{E}\sum_{j=1}^{d}(\sigma_{l,j}^{2}+6K\sigma_{g,j}^{2})$$

Here we used the fact that $\sqrt{v_{t-1,j}}\leq\eta_{l}KG\sqrt{T}$ and Condition I in Theorem 1. Using the above bound in Equation 5, we get

$$T_{1} \leq -\frac{\eta_{l}K}{4}\sum_{j=1}^{d}\frac{[\nabla f(x_{t})]_{j}^{2}}{\sqrt{v_{t-1,j}}+\tau}+\frac{5\eta_{l}^{3}K^{2}L^{2}}{2\tau}\mathbb{E}\sum_{j=1}^{d}(\sigma_{l,j}^{2}+6K\sigma_{g,j}^{2}) \qquad (8)$$

**Putting the pieces together** Substituting in Equation (3), bounds $T_1$ in Equation (8) and bound $T_2$ in Equation (4), we obtain

$$\mathbb{E}_{t}[f(x_{t+1})] \leq f(x_{t})+\eta\times\left[-\frac{\eta_{l}K}{4}\sum_{j=1}^{d}\frac{[\nabla f(x_{t})]_{j}^{2}}{\sqrt{v_{t-1,j}}+\tau}+\frac{5\eta_{l}^{3}K^{2}L^{2}}{2\tau}\mathbb{E}\sum_{j=1}^{d}(\sigma_{l,j}^{2}+6K\sigma_{g,j}^{2})\right]$$

$$+\eta\times\mathbb{E}\sum_{j=1}^{d}\frac{\eta_{l}KG^{2}}{\tau}\left[\frac{\Delta_{t,j}^{2}}{\sum_{l=0}^{t}\Delta_{l,j}^{2}+\tau^{2}}\right]+\frac{\eta^{2}}{2}\sum_{j=1}^{d}L\mathbb{E}\left[\frac{\Delta_{t,j}^{2}}{\sum_{l=0}^{t}\Delta_{l,j}^{2}+\tau^{2}}\right]. \qquad (9)$$

Rearranging the above inequality and summing it from $t=0$ to $T-1$, we get

$$\sum_{t=0}^{T-1}\frac{\eta_{l}K}{4}\sum_{j=1}^{d}\mathbb{E}\frac{[\nabla f(x_{t})]_{j}^{2}}{\sqrt{v_{t-1,j}}+\tau} \leq \frac{f(x_{0})-\mathbb{E}[f(x_{T})]}{\eta}+\frac{5\eta_{l}^{3}K^{2}L^{2}T}{2\tau}\sum_{j=1}^{d}(\sigma_{l,j}^{2}+6K\sigma_{g,j}^{2})$$

$$+\sum_{t=0}^{T-1}\mathbb{E}\sum_{j=1}^{d}\left(\frac{\eta_{l}KG^{2}}{\tau}+\frac{\eta L}{2}\right)\times\left[\frac{\Delta_{t,j}^{2}}{\sum_{l=0}^{t}\Delta_{l,j}^{2}+\tau^{2}}\right] \qquad (10)$$

The first inequality uses simple telescoping sum. For completing the proof, we need the following result.

**Lemma 4.** *The following upper bound holds for Algorithm 2 (*FEDADAGRAD*):*

$$\mathbb{E}\sum_{t=0}^{T-1}\sum_{j=1}^{d}\frac{\Delta_{t,j}^2}{\sum_{l=0}^{t}\Delta_{l,j}^2 + \tau^2} \le \left[\min\left\{d + \sum_{j=1}^{d}\log\left(1 + \frac{\eta_l^2 K^2 G^2 T}{\tau^2}\right)\right.\right.$$
$$\left.\left. + \frac{4\eta_l^2 KT}{m\tau^2}\sum_{j=1}^{d}\sigma_{l,j}^2 + 20\eta_l^4 K^3 L^2 T\mathbb{E}\sum_{j=1}^{d}\frac{(\sigma_{l,j}^2 + 6K\sigma_{g,j}^2)}{\tau^2} + \frac{40\eta_l^4 K^2 L^2}{\tau^2}\sum_{t=0}^{T-1}\mathbb{E}\left[\|\nabla f(x_t)\|^2\right]\right\}\right]$$

*Proof.* We bound the desired quantity in the following manner:

$$\mathbb{E}\sum_{t=0}^{T-1}\sum_{j=1}^{d}\frac{\Delta_{t,j}^2}{\sum_{l=0}^{t}\Delta_{l,j}^2 + \tau^2} \le d + \mathbb{E}\sum_{j=1}^{d}\log\left(1 + \frac{\sum_{l=0}^{T-1}\Delta_{l,j}^2}{\tau^2}\right) \le d + \sum_{j=1}^{d}\log\left(1 + \frac{\eta_l^2 K^2 G^2 T}{\tau^2}\right).$$

An alternate way of the bounding this quantity is as follows:

$$\mathbb{E}\sum_{t=0}^{T-1}\sum_{j=1}^{d}\frac{\Delta_{t,j}^2}{\sum_{l=0}^{t}\Delta_{t,j}^2 + \tau^2} \le \mathbb{E}\sum_{t=0}^{T-1}\sum_{j=1}^{d}\frac{\Delta_{t,j}^2}{\tau^2}$$
$$\le \mathbb{E}\sum_{t=0}^{T-1}\left\|\frac{\Delta_t + \eta_l K\nabla f(x_t) - \eta_l K\nabla f(x_t)}{\tau}\right\|^2$$
$$\le 2\mathbb{E}\sum_{t=0}^{T-1}\left[\left\|\frac{\Delta_t + \eta_l K\nabla f(x_t)}{\tau}\right\|^2 + \eta_l^2 K^2\left\|\frac{\nabla f(x_t)}{\tau}\right\|^2\right]. \tag{11}$$

The first quantity in the above bound can be further bounded as follows:

$$2\mathbb{E}\sum_{t=0}^{T-1}\left\|\frac{1}{\tau}\cdot\left(-\frac{1}{m}\sum_{i=1}^{m}\sum_{k=0}^{K-1}\eta_l g_{i,k}^t + \eta_l K\nabla f(x_t)\right)\right\|^2$$
$$= 2\mathbb{E}\sum_{t=0}^{T-1}\left[\left\|\frac{1}{\tau}\cdot\left(\frac{1}{m}\sum_{i=1}^{m}\sum_{k=0}^{K-1}\left(\eta_l g_{i,k}^t - \eta_l\nabla F_i(x_{i,k}^t) + \eta_l\nabla F_i(x_{i,k}^t) - \eta_l\nabla F_i(x_t) + \eta_l\nabla F_i(x_t)\right) - \eta_l K\nabla f(x_t)\right)\right\|^2\right]$$
$$= \frac{2\eta_l^2}{m^2}\sum_{t=0}^{T-1}\mathbb{E}\left[\left\|\sum_{i=1}^{m}\sum_{k=0}^{K-1}\frac{1}{\tau}\cdot\left(g_{i,k}^t - \nabla F_i(x_{i,k}^t) + \nabla F_i(x_{i,k}^t) - \nabla F_i(x_t)\right)\right\|^2\right]$$
$$\le \frac{4\eta_l^2}{m^2}\sum_{t=0}^{T-1}\mathbb{E}\left[\left\|\sum_{i=1}^{m}\sum_{k=0}^{K-1}\frac{1}{\tau}\cdot\left(g_{i,k}^t - \nabla F_i(x_{i,k}^t)\right)\right\|^2 + \left\|\sum_{i=1}^{m}\sum_{k=0}^{K-1}\frac{1}{\tau}\cdot\left(\nabla F_i(x_{i,k}^t) - \nabla F_i(x_t)\right)\right\|^2\right]$$
$$\le \frac{4\eta_l^2 KT}{m}\sum_{j=1}^{d}\frac{\sigma_{l,j}^2}{\tau^2} + \frac{4\eta_l^2 K}{m}\mathbb{E}\sum_{i=1}^{m}\sum_{k=0}^{K-1}\sum_{t=0}^{T-1}\left\|\frac{1}{\tau}\cdot\left(\nabla F_i(x_{i,k}^t) - \nabla F_i(x_t)\right)\right\|^2$$
$$\le \frac{4\eta_l^2 KT}{m}\sum_{j=1}^{d}\frac{\sigma_{l,j}^2}{\tau^2} + \frac{4\eta_l^2 K}{m}\mathbb{E}\sum_{i=1}^{m}\sum_{k=0}^{K-1}\sum_{t=0}^{T-1}\left\|\frac{L}{\tau}\cdot\left(x_{i,k}^t - x_t\right)\right\|^2 \quad \text{(by Assumptions 1 and 2)}$$
$$\le \frac{4\eta_l^2 KT}{m\tau^2}\sum_{j=1}^{d}\sigma_{l,j}^2 + 20\eta_l^4 K^3 L^2 T\sum_{j=1}^{d}\frac{(\sigma_{l,j}^2 + 6K\sigma_{g,j}^2)}{\tau^2} + \frac{40\eta_l^4 K^4 L^2}{\tau^2}\sum_{t=0}^{T-1}\mathbb{E}\left[\|\nabla f(x_t)\|^2\right] \quad \text{(by Lemma 3)}.$$

Here, the first inequality follows from simple application of the fact that $ab \le (a^2 + b^2)/2$. The result follows. □

Substituting the above bound in Equation (10), we obtain:

$$\frac{\eta_l K}{4} \sum_{t=0}^{T-1} \sum_{j=1}^{d} \mathbb{E} \frac{[\nabla f(x_t)]_j^2}{\sqrt{v_{t-1,j}} + \tau}$$

$$\leq \frac{f(x_0) - \mathbb{E}[f(x_T)]}{\eta} + \frac{5\eta_l^3 K^2 L^2}{2\tau} \mathbb{E} \sum_{t=0}^{T-1} \sum_{j=1}^{d} (\sigma_{l,j}^2 + 6K\sigma_{g,j}^2)$$

$$+ \sum_{j=1}^{d} \left( \frac{\eta_l K G^2}{\tau} + \frac{\eta L}{2} \right) \times \left[ \min \left\{ d + d \log \left( 1 + \frac{\eta_l^2 K^2 G^2 T}{\tau^2} \right) + \right.\right.$$

$$\left.\left. \frac{4\eta_l^2 KT}{m\tau^2} \sum_{j=1}^{d} \sigma_{l,j}^2 + 20\eta_l^4 K^3 L^2 T \sum_{j=1}^{d} \frac{(\sigma_{l,j}^2 + 6K\sigma_{g,j}^2)}{\tau^2} + \frac{40\eta_l^4 K^4 L^2}{\tau^2} \sum_{t=0}^{T-1} \mathbb{E}\left[\|\nabla f(x_t)\|^2\right] \right\} \right]$$

$$(12)$$

We observe the following:

$$\sum_{t=0}^{T-1} \sum_{j=1}^{d} \frac{[\nabla \mathbb{E} f(x_t)]_j^2}{\sqrt{v_{t-1,j}} + \tau} \geq \sum_{t=0}^{T-1} \sum_{j=1}^{d} \mathbb{E} \frac{[\nabla f(x_t)]_j^2}{\eta_l K G \sqrt{T} + \tau} \geq \frac{T}{\eta_l K G \sqrt{T} + \tau} \min_{0 \leq t \leq T} \mathbb{E}\|\nabla f(x_t)\|^2.$$

The second part of Theorem 1 follows from using the above inequality in Equation (12). Note that the first part of Theorem 1 is obtain from the part of Lemma 4. □

### A.2.1 LIMITED PARTICIPATION

For limited participation, the main changed in the proof is in Equation (11). The rest of the proof is similar so we mainly focus on Equation (11) here. Let $\mathcal{S}$ be the sampled set at the $t^{\text{th}}$ iteration such that $|\mathcal{S}| = s$. In partial participation, we assume that the set $\mathcal{S}$ is sampled uniformly from all subsets of $[m]$ with size $s$. In this case, for the first term in Equation (11), we have

$$2\mathbb{E} \sum_{t=0}^{T-1} \left\| \frac{1}{\tau} \cdot \left( -\frac{1}{|\mathcal{S}|} \sum_{i \in \mathcal{S}} \sum_{k=0}^{K-1} \eta_l g_{i,k}^t + \eta_l K \nabla f(x_t) \right) \right\|^2$$

$$= 2\mathbb{E} \sum_{t=0}^{T-1} \left[ \left\| \frac{1}{\tau} \cdot \left( \frac{1}{s} \sum_{i \in \mathcal{S}} \sum_{k=0}^{K-1} \left( \eta_l g_{i,k}^t - \eta_l \nabla F_i(x_{i,k}^t) + \eta_l \nabla F_i(x_{i,k}^t) - \eta_l \nabla F_i(x_t) + \eta_l \nabla F_i(x_t) \right) - \eta_l K \nabla f(x_t) \right) \right\|^2 \right]$$

$$\leq \frac{6\eta_l^2}{s^2} \sum_{t=0}^{T-1} \mathbb{E} \left[ \left\| \sum_{i \in \mathcal{S}} \sum_{k=0}^{K-1} \frac{1}{\tau} \cdot \left( g_{i,k}^t - \nabla F_i(x_{i,k}^t) \right) \right\|^2 + \left\| \sum_{i \in \mathcal{S}} \sum_{k=0}^{K-1} \frac{1}{\tau} \cdot \left( \nabla F_i(x_{i,k}^t) - \nabla F_i(x_t) \right) \right\|^2 \right.$$

$$\left. + K^2 \left\| \frac{1}{\tau} \cdot \left( \sum_{i \in \mathcal{S}} F_i(x_t) - s \nabla f(x_t) \right) \right\|^2 \right]$$

$$\leq \frac{6\eta_l^2 KT}{s} \sum_{j=1}^{d} \frac{\sigma_{l,j}^2}{\tau^2} + \frac{6\eta_l^2 K}{s} \mathbb{E} \sum_{i \in \mathcal{S}} \sum_{k=0}^{K-1} \sum_{t=0}^{T-1} \left\| \frac{1}{\tau} \cdot \left( \nabla F_i(x_{i,k}^t) - \nabla F_i(x_t) \right) \right\|^2 + \frac{6\eta_l^2 K^2 T \sigma_g^2}{\tau^2} \left( 1 - \frac{s}{m} \right)$$

$$\leq \frac{6\eta_l^2 KT}{s} \sum_{j=1}^{d} \frac{\sigma_{l,j}^2}{\tau^2} + \frac{6\eta_l^2 K}{s} \mathbb{E} \sum_{i \in \mathcal{S}} \sum_{k=0}^{K-1} \sum_{t=0}^{T-1} \left\| \frac{L}{\tau} \cdot \left( x_{i,k}^t - x_t \right) \right\|^2 + \frac{6\eta_l^2 K^2 T \sigma_g^2}{\tau^2} \left( 1 - \frac{s}{m} \right)$$

$$\leq \frac{6\eta_l^2 KT}{\tau^2 s} \sum_{j=1}^{d} \sigma_{l,j}^2 + 30\eta_l^4 K^3 L^2 T \mathbb{E} \sum_{j=1}^{d} \frac{(\sigma_{l,j}^2 + 6K\sigma_{g,j}^2)}{\tau^2} + \frac{60\eta_l^4 K^4 L^2}{\tau^2} \sum_{t=0}^{T-1} \mathbb{E}\left[\|\nabla f(x_t)\|^2\right]$$

$$+ \frac{6\eta_l^2 K^2 T \sigma_g^2}{\tau^2} \left( 1 - \frac{s}{m} \right).$$

Note that the expectation here also includes $\mathcal{S}$. The first inequality is obtained from the fact that $(a + b + c)^2 \leq 3(a^2 + b^2 + c^2)$. The second inequality is obtained from the fact that set $\mathcal{S}$ is uniformly

sampled from all subsets of $[m]$ with size $s$. The third and fourth inequalities are similar to the one used in proof of Theorem 1. Substituting the above bound in Equation (10) gives the desired convergence rate.

### A.3 PROOF OF THEOREM 2

*Proof of Theorem 2.* The proof strategy is similar to that of FEDADAGRAD except that we need to handle the exponential moving average in FEDADAM. We note that the update of FEDADAM is the following

$$x_{t+1} = x_t + \eta \frac{\Delta_t}{\sqrt{v_t} + \tau},$$

for all $i \in [d]$. Using the $L$-smooth nature of function $f$ and the above update rule, we have the following:

$$f(x_{t+1}) \le f(x_t) + \eta \left\langle \nabla f(x_t), \frac{\Delta_t}{\sqrt{v_t} + \tau} \right\rangle + \frac{\eta^2 L}{2} \sum_{i=1}^{d} \frac{\Delta_{t,i}^2}{(\sqrt{v_{t,i}} + \tau)^2} \qquad (13)$$

The second step follows simply from FEDADAM's update. We take the expectation of $f(x_{t+1})$ (over randomness at time step $t$) and rewrite the above inequality as:

$$\mathbb{E}_t[f(x_{t+1})] \le f(x_t) + \eta \left\langle \nabla f(x_t), \mathbb{E}_t \left[ \frac{\Delta_t}{\sqrt{v_t} + \tau} - \frac{\Delta_t}{\sqrt{\beta_2 v_{t-1}} + \tau} + \frac{\Delta_t}{\sqrt{\beta_2 v_{t-1}} + \tau} \right] \right\rangle + \frac{\eta^2 L}{2} \sum_{j=1}^{d} \mathbb{E}_t \left[ \frac{\Delta_{t,j}^2}{(\sqrt{v_{t,j}} + \tau)^2} \right]$$

$$= f(x_t) + \eta \underbrace{\left\langle \nabla f(x_t), \mathbb{E}_t \left[ \frac{\Delta_t}{\sqrt{\beta_2 v_{t-1}} + \tau} \right] \right\rangle}_{R_1} + \eta \underbrace{\left\langle \nabla f(x_t), \mathbb{E}_t \left[ \frac{\Delta_t}{\sqrt{v_t} + \tau} - \frac{\Delta_t}{\sqrt{\beta_2 v_{t-1}} + \tau} \right] \right\rangle}_{R_2}$$

$$+ \frac{\eta^2 L}{2} \sum_{j=1}^{d} \mathbb{E}_t \left[ \frac{\Delta_{t,j}^2}{(\sqrt{v_{t,j}} + \tau)^2} \right] \qquad (14)$$

**Bounding $R_2$.** We observe the following about $R_2$:

$$R_2 = \mathbb{E}_t \sum_{j=1}^{d} [\nabla f(x_t)]_j \times \left[ \frac{\Delta_{t,j}}{\sqrt{v_{t,j}} + \tau} - \frac{\Delta_{t,j}}{\sqrt{\beta_2 v_{t-1,j}} + \tau} \right]$$

$$= \mathbb{E}_t \sum_{j=1}^{d} [\nabla f(x_t)]_j \times \Delta_{t,j} \times \left[ \frac{\sqrt{\beta_2 v_{t-1,j}} - \sqrt{v_{t,j}}}{(\sqrt{v_{t,j}} + \tau)(\sqrt{\beta_2 v_{t-1,j}} + \tau)} \right]$$

$$= \mathbb{E}_t \sum_{j=1}^{d} [\nabla f(x_t)]_j \times \Delta_{t,j} \times \left[ \frac{-(1 - \beta_2)\Delta_{t,j}^2}{(\sqrt{v_{t,j}} + \tau)(\sqrt{\beta_2 v_{t-1,j}} + \tau)(\sqrt{\beta_2 v_{t-1,j}} + \sqrt{v_{t,j}})} \right]$$

$$\le (1 - \beta_2) \mathbb{E}_t \sum_{j=1}^{d} |\nabla f(x_t)]_j| \times |\Delta_{t,j}| \times \left[ \frac{\Delta_{t,j}^2}{(\sqrt{v_{t,j}} + \tau)(\sqrt{\beta_2 v_{t-1,j}} + \tau)(\sqrt{\beta_2 v_{t-1,j}} + \sqrt{v_{t,j}})} \right]$$

$$\le \sqrt{1 - \beta_2} \mathbb{E}_t \sum_{j=1}^{d} |\nabla f(x_t)]_j| \times \left[ \frac{\Delta_{t,j}^2}{\sqrt{v_{t,j}} + \tau)(\sqrt{\beta_2 v_{t-1,j}} + \tau)} \right]$$

$$\le \sqrt{1 - \beta_2} \mathbb{E}_t \sum_{j=1}^{d} \frac{G}{\tau} \times \left[ \frac{\Delta_{t,j}^2}{\sqrt{v_{t,j}} + \tau} \right].$$

**Bounding $R_1$.** The term $R_1$ can be bounded as follows:

$$
R_1 = \left\langle \nabla f(x_t), \mathbb{E}_t \left[ \frac{\Delta_t}{\sqrt{\beta_2 v_{t-1}} + \tau} \right] \right\rangle
$$

$$
= \left\langle \frac{\nabla f(x_t)}{\sqrt{\beta_2 v_{t-1}} + \tau}, \mathbb{E}_t \left[ \Delta_t - \eta_l K \nabla f(x_t) + \eta_l K \nabla f(x_t) \right] \right\rangle
$$

$$
= -\eta_l K \sum_{j=1}^{d} \frac{[\nabla f(x_t)]_j^2}{\sqrt{\beta_2 v_{t-1,j}} + \tau} + \underbrace{\left\langle \frac{\nabla f(x_t)}{\sqrt{\beta_2 v_{t-1}} + \tau}, \mathbb{E}_t \left[ \Delta_t + \eta_l K \nabla f(x_t) \right] \right\rangle}_{R_3}. \tag{15}
$$

**Bounding $R_3$.** The term $R_3$ can be bounded in exactly the same way as term $T_3$ in proof of Theorem 1:

$$
R_3 \leq \frac{\eta_l K}{2} \sum_{j=1}^{d} \frac{[\nabla f(x_t)]_j^2}{\sqrt{\beta_2 v_{t-1,j}} + \tau} + \frac{\eta_l L^2}{2m\tau} \mathbb{E}_t \left[ \sum_{i=1}^{m} \sum_{k=0}^{K-1} \|x_{i,k}^t - x_t\|^2 \right]
$$

Substituting the above inequality in Equation (15), we get

$$
R_1 \leq -\frac{\eta_l K}{2} \sum_{j=1}^{d} \frac{[\nabla f(x_t)]_j^2}{\sqrt{\beta_2 v_{t-1,j}} + \tau} + \frac{\eta_l L^2}{2m\tau} \mathbb{E}_t \left[ \sum_{i=1}^{m} \sum_{k=0}^{K-1} \|x_{i,k}^t - x_t\|^2 \right]
$$

Here we used the fact that $\sqrt{v_{t-1,j}} \leq \eta_l K G$ and conditions in Theorem 2. Using Lemma 3, we obtain the following bound on $R_1$:

$$
R_1 \leq -\frac{\eta_l K}{4} \sum_{j=1}^{d} \frac{[\nabla f(x_t)]_j^2}{\sqrt{\beta_2 v_{t-1,j}} + \tau} + \frac{5\eta_l^3 K^2 L^2}{2\tau} \mathbb{E}_t \sum_{j=1}^{d} (\sigma_{l,j}^2 + 6K\sigma_{g,j}^2) \tag{16}
$$

**Putting pieces together.** Substituting bounds $R_1$ and $R_2$ in Equation (14), we have

$$
\mathbb{E}_t[f(x_{t+1})] \leq f(x_t) - \frac{\eta \eta_l K}{4} \sum_{j=1}^{d} \frac{[\nabla f(x_t)]_j^2}{\sqrt{\beta_2 v_{t-1,j}} + \tau} + \frac{5\eta \eta_l^3 K^2 L^2}{2\tau} \mathbb{E} \sum_{j=1}^{d} (\sigma_{l,j}^2 + 6K\sigma_{g,j}^2)
$$

$$
+ \left( \frac{\eta \sqrt{1-\beta_2} G}{\tau} \right) \sum_{j=1}^{d} \mathbb{E}_t \left[ \frac{\Delta_{t,j}^2}{\sqrt{v_{t,j}} + \tau)} \right] + \left( \frac{\eta^2 L}{2} \right) \sum_{j=1}^{d} \mathbb{E}_t \left[ \frac{\Delta_{t,j}^2}{v_{t,j} + \tau^2)} \right]
$$

Summing over $t = 0$ to $T - 1$ and using telescoping sum, we have

$$
\mathbb{E}[f(x_T)] \leq f(x_0) - \frac{\eta \eta_l K}{4} \sum_{t=0}^{T-1} \sum_{j=1}^{d} \mathbb{E} \frac{[\nabla f(x_t)]_j^2}{\sqrt{\beta_2 v_{t-1,j}} + \tau} + \frac{5\eta \eta_l^3 K^2 L^2 T}{2\tau} \mathbb{E} \sum_{j=1}^{d} (\sigma_{l,j}^2 + 6K\sigma_{g,j}^2)
$$

$$
\left( \frac{\eta \sqrt{1-\beta_2} G}{\tau} \right) \sum_{t=0}^{T-1} \sum_{j=1}^{d} \mathbb{E} \left[ \frac{\Delta_{t,j}^2}{\sqrt{v_{t,j}} + \tau)} \right] + \left( \frac{\eta^2 L}{2} \right) \sum_{t=0}^{T-1} \sum_{j=1}^{d} \mathbb{E} \left[ \frac{\Delta_{t,j}^2}{v_{t,j} + \tau^2)} \right] \tag{17}
$$

To bound this term further, we need the following result.

**Lemma 5.** *The following upper bound holds for Algorithm 2 (*FEDADAM*):*

$$
\sum_{t=0}^{T-1} \sum_{j=1}^{d} \mathbb{E} \left[ \frac{\Delta_{t,j}^2}{(v_{t,j} + \tau^2)} \right] \leq \frac{4\eta_l^2 KT}{m\tau^2} \sum_{j=1}^{d} \sigma_{l,j}^2 + \frac{20\eta_l^4 K^3 L^2 T}{\tau^2} \mathbb{E} \sum_{j=1}^{d} (\sigma_{l,j}^2 + 6K\sigma_{g,j}^2) + \frac{40\eta_l^4 K^4 L^2}{\tau^2} \sum_{t=0}^{T-1} \mathbb{E} \left[ \|\nabla f(x_t)\|^2 \right]
$$

*Proof.*

$$
\mathbb{E} \sum_{t=0}^{T-1} \sum_{j=1}^{d} \frac{\Delta_{t,j}^2}{(1-\beta_2) \sum_{l=0}^{t} \beta_2^{t-l} \Delta_{t,j}^2 + \tau^2} \leq \mathbb{E} \sum_{t=0}^{T-1} \sum_{j=1}^{d} \frac{\Delta_{t,j}^2}{\tau^2}
$$

The rest of the proof follows along the lines of proof of Lemma 4. Using the same argument, we get

$$\sum_{j=1}^{d} \mathbb{E}\left[\frac{\Delta_{t,j}^2}{(v_{t,j}+\tau^2)}\right] \leq \frac{4\eta_l^2 KT}{m\tau^2}\sum_{j=1}^{d}\sigma_{l,j}^2 + \frac{20\eta_l^4 K^3 L^2 T}{\tau^2}\mathbb{E}\sum_{j=1}^{d}(\sigma_{l,j}^2 + 6K\sigma_{g,j}^2)$$
$$+ \frac{40\eta_l^4 K^4 L^2}{\tau^2}\sum_{t=0}^{T-1}\mathbb{E}\left[\|\nabla f(x_t)\|^2\right],$$

which is the desired result. □

Substituting the bound obtained from above lemma in Equation (17) and using a similar argument for bounding

$$\left(\frac{\eta\sqrt{1-\beta_2}G}{\tau}\right)\sum_{t=0}^{T-1}\sum_{j=1}^{d}\mathbb{E}\left[\frac{\Delta_{t,j}^2}{\sqrt{v_{t,j}}+\tau)}\right],$$

we obtain

$$\mathbb{E}_t[f(x_T)] \leq f(x_0) - \frac{\eta\eta_l K}{8}\sum_{t=0}^{T-1}\sum_{j=1}^{d}\frac{[\nabla f(x_t)]_j^2}{\sqrt{\beta_2 v_{t-1,j}}+\tau} + \frac{5\eta\eta_l^3 K^2 L^2 T}{2\tau}\mathbb{E}\sum_{j=1}^{d}(\sigma_{l,j}^2 + 6K\sigma_{g,j}^2)$$
$$+ \left(\eta\sqrt{1-\beta_2}G + \frac{\eta^2 L}{2}\right)\times\left[\frac{4\eta_l^2 KT}{m\tau^2}\sum_{j=1}^{d}\sigma_{l,j}^2 + \frac{20\eta_l^4 K^4 L^2 T}{\tau^2}\mathbb{E}\sum_{j=1}^{d}(\sigma_{l,j}^2 + 6K\sigma_{g,j}^2)\right]$$

The above inequality is obtained due to the fact that:

$$\left(\sqrt{1-\beta_2}G + \frac{\eta L}{2}\right)\frac{40\eta_l^4 K^4 L^2}{\tau^2} \leq \frac{\eta_l K}{16}\left(\frac{1}{\sqrt{\beta_2}\eta_l KG} + \frac{1}{\tau}\right).$$

The above condition follows from the condition on $\eta_l$ in Theorem 2. We also observe the following:

$$\sum_{t=0}^{T-1}\sum_{j=1}^{d}\frac{[\nabla f(x_t)]_j^2}{\sqrt{\beta_2 v_{t-1,j}}+\tau} \geq \sum_{t=0}^{T-1}\sum_{j=1}^{d}\frac{[\nabla f(x_t)]_j^2}{\sqrt{\beta_2}\eta_l KG+\tau} \geq \frac{T}{\sqrt{\beta_2}\eta_l KG+\tau}\min_{0\leq t\leq T}\|\nabla f(x_t)\|^2.$$

Substituting this bound in the above equation yields the desired result. □

### A.4 Auxiliary Lemmatta

**Lemma 6.** *For random variables $z_1, \ldots, z_r$, we have*

$$\mathbb{E}\left[\|z_1 + \ldots + z_r\|^2\right] \leq r\mathbb{E}\left[\|z_1\|^2 + \ldots + \|z_r\|^2\right].$$

**Lemma 7.** *For independent, mean 0 random variables $z_1, \ldots, z_r$, we have*

$$\mathbb{E}\left[\|z_1 + \ldots + z_r\|^2\right] = \mathbb{E}\left[\|z_1\|^2 + \ldots + \|z_r\|^2\right].$$

## B Federated Algorithms: Implementations and Practical Considerations

### B.1 FedAvg and FedOpt

In Algorithm 3, we give a simplified version of the FEDAVG algorithm by McMahan et al. (2017), that applies to the setup given in Section 2. We write $\text{SGD}_K(x_t, \eta_l, f_i)$ to denote $K$ steps of SGD using gradients $\nabla f_i(x, z)$ for $z \sim \mathcal{D}_i$ with (local) learning rate $\eta_l$, starting from $x_t$. As noted in Section 2, Algorithm 3 is the special case of Algorithm 1 where CLIENTOPT is SGD, and SERVEROPT is SGD with learning rate 1.

While Algorithms 1, 2, and 3 are useful for understanding relations between federated optimization methods, we are also interested in practical versions of these algorithms. In particular, Algorithms 1, 2, and 3 are stated in terms of a kind of 'gradient oracle', where we compute unbiased estimates

---

**Algorithm 3 Simplified FEDAVG**

---

Input: $x_0$
**for** $t = 0, \cdots, T - 1$ **do**
    Sample a subset $\mathcal{S}$ of clients
    $x_i^t = x_t$
    **for** each client $i \in \mathcal{S}$ **in parallel do**
        $x_i^t = \text{SGD}_K(x_t, \eta_l, f_i)$ for $i \in \mathcal{S}$ (in parallel)
    $x_{t+1} = \frac{1}{|\mathcal{S}|} \sum_{i \in \mathcal{S}} x_i^t$

---

of the client's gradient. In practical scenarios, we often only have access to finite data samples, the number of which may vary between clients.

Instead, we assume that in (1), each client distribution $\mathcal{D}_i$ is the uniform distribution over some finite set $D_i$ of size $n_i$. The $n_i$ may vary significantly between clients, requiring extra care when implementing federated optimization methods. We assume the set $D_i$ is partitioned into a collection of batches $\mathcal{B}_i$, each of size $B$. For $b \in \mathcal{B}_i$, we let $f_i(x; b)$ denote the average loss on this batch at $x$ with corresponding gradient $\nabla f_i(x; b)$. Thus, if $b$ is sampled uniformly at random from $\mathcal{B}_i$, $\nabla f_i(x; b)$ is an unbiased estimate of $\nabla F_i(x)$.

When training, instead of uniformly using $K$ gradient steps, as in Algorithm 1, we will instead perform $E$ epochs of training over each client's dataset. Additionally, we will take a weighted average of the client updates, where we weight according to the number of examples $n_i$ in each client's dataset. This leads to a batched data version of FEDOPT in Algorithm 4, and a batched data version of FEDADAGRAD, FEDADAM, and FEDYOGI given in Algorithm 5.

---

**Algorithm 4 FEDOPT** - Batched data

---

Input: $x_0$, CLIENTOPT, SERVEROPT
**for** $t = 0, \cdots, T - 1$ **do**
    Sample a subset $\mathcal{S}$ of clients
    $x_i^t = x_t$
    **for** each client $i \in \mathcal{S}$ **in parallel do**
        **for** $e = 1, \ldots, E$ **do**
            **for** $b \in \mathcal{B}_i$ **do**
                $g_i^t = \nabla f_i(x_i^t; b)$
                $x_i^t = \text{CLIENTOPT}(x_i^t, g_i^t, \eta_l, t)$
        $\Delta_i^t = x_i^t - x_t$
    $n = \sum_{i \in \mathcal{S}} n_i, \ \Delta_t = \sum_{i \in \mathcal{S}} \frac{n_i}{n} \Delta_i^t$
    $x_{t+1} = \text{SERVEROPT}(x_t, -\Delta_t, \eta, t)$

---

In Section 5, we use Algorithm 4 when implementing FEDAVG and FEDAVGM. In particular, FEDAVG and FEDAVGM correspond to Algorithm 4 where CLIENTOPT and SERVEROPT are SGD. FEDAVG uses vanilla SGD on the server, while FEDAVGM uses SGD with a momentum parameter of 0.9. In both methods, we tune both client learning rate $\eta_l$ and server learning rate $\eta$. This means that the version of FEDAVG used in all experiments is strictly more general than that in (McMahan et al., 2017), which corresponds to FEDOPT where CLIENTOPT and SERVEROPT are SGD, and SERVEROPT has a learning rate of 1.

We use Algorithm 5 for all implementations FEDADAGRAD, FEDADAM, and FEDYOGI in Section 5. For FEDADAGRAD, we set $\beta_1 = \beta_2 = 0$ (as typical versions of ADAGRAD do not use momentum). For FEDADAM and FEDYOGI we set $\beta_1 = 0.9, \beta_2 = 0.99$. While these parameters are generally good choices (Zaheer et al., 2018), we emphasize that better results may be obtainable by tuning these parameters.

### B.2 SCAFFOLD

As discussed in Section 5, we compare all five optimizers above to SCAFFOLD (Karimireddy et al., 2019) on our various tasks. There are a few important notes about the validity of this comparison.

---

**Algorithm 5** FEDADAGRAD , FEDYOGI , and FEDADAM - Batched data

---

Input: $x_0, v_{-1} \geq \tau^2$, optional $\beta_1, \beta_2 \in [0, 1)$ for FEDYOGI and FEDADAM
**for** $t = 0, \cdots, T - 1$ **do**
    Sample a subset $\mathcal{S}$ of clients
    $x_i^t = x_t$
    **for** each client $i \in \mathcal{S}$ **in parallel do**
        **for** $e = 1, \ldots, E$ **do**
            **for** $b \in \mathcal{B}_i$ **do**
                $x_i^t = x_i^t - \eta_l \nabla f_i(x_i^t; b)$
        $\Delta_i^t = x_i^t - x_t$
    $n = \sum_{i \in \mathcal{S}} n_i, \ \Delta_t = \sum_{i \in \mathcal{S}} \frac{n_i}{n} \Delta_i^t$
    $\Delta_t = \beta_1 \Delta_{t-1} + (1 - \beta_1) \Delta_t$
    $v_t = v_{t-1} + \Delta_t^2$ (**FEDADAGRAD**)
    $v_t = v_{t-1} - (1 - \beta_2) \Delta_t^2 \, \text{sign}(v_{t-1} - \Delta_t^2)$ (**FEDYOGI**)
    $v_t = \beta_2 v_{t-1} + (1 - \beta_2) \Delta_t^2$ (**FEDADAM**)
    $x_{t+1} = x_t + \eta \frac{\Delta_t}{\sqrt{v_t} + \tau}$

---

1. In cross-device settings, this is not a fair comparison. In particular, SCAFFOLD does not work in settings where clients cannot maintain state across rounds, as may be the case for federated learning systems on edge devices, such as cell phones.

2. SCAFFOLD has two variants described by Karimireddy et al. (2019). In Option I, the *control variate* of a client is updated using a full gradient computation. This effectively requires performing an extra pass over each client's dataset, as compared to Algorithm 1. In order to normalize the amount of client work, we instead use Option II, in which the clients' control variates are updated using the difference between the server model and the client's learned model. This requires the same amount of client work as FEDAVG and Algorithm 2.

For practical reasons, we implement a version of SCAFFOLD mirroring Algorithm 4, in which we perform $E$ epochs over the client's dataset, and perform *weighted averaging* of client models. For posterity, we give the full pseudo-code of the version of SCAFFOLD used in our experiments in Algorithm 6. This is a simple adaptiation of Option II of the SCAFFOLD algorithm in (Karimireddy et al., 2019) to the same setting as Algorithm 4. In particular, we let $n_i$ denote the number of examples in client $i$'s local dataset.

---

**Algorithm 6 SCAFFOLD**, Option II - Batched data

---

Input: $x_0, c, \eta_l, \eta$
**for** $t = 0, \cdots, T - 1$ **do**
    Sample a subset $\mathcal{S}$ of clients
    $x_i^t = x_t$
    **for** each client $i \in \mathcal{S}$ **in parallel do**
        **for** $e = 1, \ldots, E$ **do**
            **for** $b \in \mathcal{B}_i$ **do**
                $g_i^t = \nabla f_i(x_i^t; b)$
                $x_i^t = x_i^t - \eta_l(g_i^t - c_i + c)$
        $c_i^+ = c_i - c + (E|\mathcal{B}_i|\eta_l)^{-1}(x_i^t - x_i)$
        $\Delta x_i = x_i^t - x_t, \Delta c_i = c_i^+ - c_i$
        $c_i = c_i^+$
    $n = \sum_{i \in \mathcal{S}} n_i, \ \Delta x = \sum_{i \in \mathcal{S}} \frac{n_i}{n} \Delta x_i, \ \Delta c = \sum_{i \in \mathcal{S}} \frac{n_i}{n} \Delta c_i$
    $x_{t+1} = x_t + \eta \Delta x, \ c = c + \frac{|\mathcal{S}|}{N} \Delta c$

---

In this algorithm, $c_i$ is the *control variate* of client $i$, and $c$ is the running average of these control variates. In practice, we must initialize the control variates $c_i$ in some way when sampling a client $i$ for the first time. In our implementation, we set $c_i = c$ the first time we sample a client $i$. This has

the advantage of exactly recovering FEDAVG when each client is sampled at most once. To initialize $c$, we use the all zeros vector.

We compare this version of SCAFFOLD to FEDADAGRAD, FEDADAM, FEDYOGI, FEDAVGM, and FEDAVG on our tasks, tuning the learning rates in the same way (using the same grids as in Appendix D.2). In particular, $\eta_l, \eta$ are tuned to obtain the best training performance over the last 100 communication rounds. We use the same federated hyperparameters for SCAFFOLD as discussed in Section 4. Namely, we set $E = 1$, and sample 10 clients per round for all tasks except Stack Overflow NWP, where we sample 50. The results are given in Figure 1 in Section 5.

### B.3 LOOKAHEAD, ADAALTER, AND CLIENT ADAPTIVITY

The LOOKAHEAD optimizer (Zhang et al., 2019b) is primarily designed for non-FL settings. LOOKAHEAD uses a generic optimizer in the inner loop and updates its parameters using a "outer" learning rate. Thus, unlike FEDOPT, LOOKAHEAD uses a single generic optimizer and is thus conceptually different. In fact, LOOKAHEAD can be seen as a special case of FEDOPT in non-FL settings which uses a generic optimizer CLIENTOPT as a client optimizer, and SGD as the server optimizer. While there are multiple ways LOOKAHEAD could be generalized to a federated setting, one straightforward version would simply use an adaptive method as the CLIENTOPT. On the other hand, ADAALTER (Xie et al., 2019) is designed specifically for distributed settings. In ADAALTER, clients use a local optimizer similar to ADAGRAD (McMahan & Streeter, 2010a; Duchi et al., 2011) to perform multiple epochs of training on their local datasets. Both LOOKAHEAD and ADAALTER use client adaptivity, which is fundamentally different from the adaptive *server* optimizers proposed in Algorithm 2.

To illustrate the differences, consider the client-to-server communication in ADAALTER. This requires communicating both the model weights and the client accumulators (used to perform the adaptive optimization, analogous to $v_t$ in Algorithm 2). In the case of ADAALTER, the client accumulator is the same size as the model's trainable weights. Thus, the client-to-server communication doubles for this method, relative to FEDAVG. In ADAALTER, the server averages the client accumulators and broadcasts the average to the next set of clients, who use this to initialize their adaptive optimizers. This means that the server-to-client communication also doubles relative to FEDAVG. The same would occur for any adaptive client optimizer in the distributed version of LOOKAHEAD described above. For similar reasons, we see that client adaptive methods also increase the amount of memory needed on the client (as they must store the current accumulator). By contrast, our adaptive server methods (Algorithm 2) do not require extra communication or client memory relative to FEDAVG. Thus, we see that server-side adaptive optimization benefits from lower per-round communication and client memory requirements, which are of paramount importance for FL applications (Bonawitz et al., 2019).

## C  DATASET & MODELS

Here we provide detailed description of the datasets and models used in the paper. We use federated versions of vision datasets EMNIST (Cohen et al., 2017), CIFAR-10 (Krizhevsky & Hinton, 2009), and CIFAR-100 (Krizhevsky & Hinton, 2009), and language modeling datasets Shakespeare (McMahan et al., 2017) and StackOverflow (Authors, 2019). Statistics for the training datasets can be found in Table 2. We give descriptions of the datasets, models, and tasks below. Statistics on the number of clients and examples in both the training and test splits of the datasets are given in Table 2.

Table 2: Dataset statistics.

| DATASET | TRAIN CLIENTS | TRAIN EXAMPLES | TEST CLIENTS | TEST EXAMPLES |
|---|---|---|---|---|
| CIFAR-10 | 500 | 50,000 | 100 | 10,000 |
| CIFAR-100 | 500 | 50,000 | 100 | 10,000 |
| EMNIST-62 | 3,400 | 671,585 | 3,400 | 77,483 |
| SHAKESPEARE | 715 | 16,068 | 715 | 2,356 |
| STACKOVERFLOW | 342,477 | 135,818,730 | 204,088 | 16,586,035 |

## C.1 CIFAR-10/CIFAR-100

We create a federated version of CIFAR-10 by randomly partitioning the training data among 500 clients, with each client receiving 100 examples. We use the same approach as Hsu et al. (2019), where we apply latent Dirichlet allocation (LDA) over the labels of CIFAR-10 to create a federated dataset. Each client has an associated multinomial distribution over labels from which its examples are drawn. The multinomial is drawn from a symmetric Dirichlet distribution with parameter 0.1.

For CIFAR-100, we perform a similar partitioning of 100 examples to 500 clients, but using a more sophisticated approach. We use a two step LDA process over the coarse and fine labels. We randomly partition the data to reflect the "coarse" and "fine" label structure of CIFAR-100 by using the Pachinko Allocation Method (PAM) (Li & McCallum, 2006). This creates more realistic client datasets, whose label distributions better resemble practical heterogeneous client datasets. We have made publicly available the specific federated version of CIFAR-100 we used for all experiments, though we avoid giving a link in this work in order to avoid de-anonymization. For complete details on how the dataset was created, see Appendix F.

We train a modified ResNet-18 on both datasets, where the batch normalization layers are replaced by group normalization layers (Wu & He, 2018). We use two groups in each group normalization layer. As shown by Hsieh et al. (2019), group normalization can lead to significant gains in accuracy over batch normalization in federated settings.

**Preprocessing** CIFAR-10 and CIFAR-100 consist of images with 3 channels of $32 \times 32$ pixels each. Each pixel is represented by an unsigned int8. As is standard with CIFAR datasets, we perform preprocessing on both training and test images. For training images, we perform a random crop to shape $(24, 24, 3)$, followed by a random horizontal flip. For testing images, we centrally crop the image to $(24, 24, 3)$. For both training and testing images, we then normalize the pixel values according to their mean and standard deviation. Namely, given an image $x$, we compute $(x - \mu)/\sigma$ where $\mu$ is the average of the pixel values in $x$, and $\sigma$ is the standard deviation.

## C.2 EMNIST

EMNIST consists of images of digits and upper and lower case English characters, with 62 total classes. The federated version of EMNIST (Caldas et al., 2018) partitions the digits by their author. The dataset has natural heterogeneity stemming from the writing style of each person. We perform two distinct tasks on EMNIST, autoencoder training (EMNIST AE) and character recognition (EMNIST CR). For EMNIST AE, we train the "MNIST" autoencoder (Zaheer et al., 2018). This is a densely connected autoencoder with layers of size $(28 \times 28) - 1000 - 500 - 250 - 30$ and a symmetric decoder. A full description of the model is in Table 3. For EMNIST CR, we use a convolutional network. The network has two convolutional layers (with $3 \times 3$ kernels), max pooling, and dropout, followed by a 128 unit dense layer. A full description of the model is in Table 4.

Table 3: EMNIST autoencoder model architecture. We use a sigmoid activation at all dense layers.

| Layer | Output Shape | # of Trainable Parameters |
|-------|--------------|---------------------------|
| Input | 784 | 0 |
| Dense | 1000 | 785000 |
| Dense | 500 | 500500 |
| Dense | 250 | 125250 |
| Dense | 30 | 7530 |
| Dense | 250 | 7750 |
| Dense | 500 | 125500 |
| Dense | 1000 | 501000 |
| Dense | 784 | 784784 |

Table 4: EMNIST character recognition model architecture.

| Layer | Output Shape | # of Trainable Parameters | Activation | Hyperparameters |
|---|---|---|---|---|
| Input | $(28, 28, 1)$ | 0 | | |
| Conv2d | $(26, 26, 32)$ | 320 | | kernel size = 3; strides=$(1, 1)$ |
| Conv2d | $(24, 24, 64)$ | 18496 | ReLU | kernel size = 3; strides=$(1, 1)$ |
| MaxPool2d | $(12, 12, 64)$ | 0 | | pool size= $(2, 2)$ |
| Dropout | $(12, 12, 64)$ | 0 | | $p = 0.25$ |
| Flatten | 9216 | 0 | | |
| Dense | 128 | 1179776 | | |
| Dropout | 128 | 0 | | $p = 0.5$ |
| Dense | 62 | 7998 | softmax | |

## C.3 SHAKESPEARE

Shakespeare is a language modeling dataset built from the collective works of William Shakespeare. In this dataset, each client corresponds to a speaking role with at least two lines. The dataset consists of 715 clients. Each client's lines are partitioned into training and test sets. Here, the task is to do next character prediction. We use an RNN that first takes a series of characters as input and embeds each of them into a learned 8-dimensional space. The embedded characters are then passed through 2 LSTM layers, each with 256 nodes, followed by a densely connected softmax output layer. We split the lines of each speaking role into into sequences of 80 characters, padding if necessary. We use a vocabulary size of 90; 86 for the characters contained in the Shakespeare dataset, and 4 extra characters for padding, out-of-vocabulary, beginning of line and end of line tokens. We train our model to take a sequence of 80 characters, and predict a sequence of 80 characters formed by shifting the input sequence by one (so that its last character is the new character we are actually trying to predict). Therefore, our output dimension is $80 \times 90$. A full description of the model is in Table 5.

Table 5: Shakespeare model architecture.

| Layer | Output Shape | # of Trainable Parameters |
|---|---|---|
| Input | 80 | 0 |
| Embedding | $(80, 8)$ | 720 |
| LSTM | $(80, 256)$ | 271360 |
| LSTM | $(80, 256)$ | 525312 |
| Dense | $(80, 90)$ | 23130 |

## C.4 STACK OVERFLOW

Stack Overflow is a language modeling dataset consisting of question and answers from the question and answer site, Stack Overflow. The questions and answers also have associated metadata, including tags. The dataset contains 342,477 unique users which we use as clients. We perform two tasks on this dataset: tag prediction via logistic regression (Stack Overflow LR, SO LR for short), and next-word prediction (Stack Overflow NWP, SO NWP for short). For both tasks, we restrict to the 10,000 most frequently used words. For Stack Overflow LR, we restrict to the 500 most frequent tags and adopt a one-versus-rest classification strategy, where each question/answer is represented as a bag-of-words vector (normalized to have sum 1).

For Stack Overflow NWP, we restrict each client to the first 1000 sentences in their dataset (if they contain this many, otherwise we use the full dataset). We also perform padding and truncation to ensure that sentences have 20 words. We then represent the sentence as a sequence of indices corresponding to the 10,000 frequently used words, as well as indices representing padding, out-of-vocabulary words, beginning of sentence, and end of sentence. We perform next-word-prediction on these sequences using an RNN that embeds each word in a sentence into a learned 96-dimensional space. It then feeds the embedded words into a single LSTM layer of hidden dimension 670, followed

by a densely connected softmax output layer. A full description of the model is in Table 6. The metric used in the main body is the top-1 accuracy over the proper 10,000-word vocabulary; it does not include padding, out-of-vocab, or beginning or end of sentence tokens when computing the accuracy.klkjlkj

Table 6: Stack Overflow next word prediction model architecture.

| Layer | Output Shape | # of Trainable Parameters |
|---|---|---|
| Input | 20 | 0 |
| Embedding | $(20, 96)$ | 960384 |
| LSTM | $(20, 670)$ | 2055560 |
| Dense | $(20, 96)$ | 64416 |
| Dense | $(20, 10004)$ | 970388 |

## D   EXPERIMENT HYPERPARAMETERS

### D.1   HYPERPARAMETER TUNING

Throughout our experiments, we compare the performance of different instantiations of Algorithm 1 that use different server optimizers. We use SGD, SGD with momentum (denoted SGDM), ADAGRAD, ADAM, and YOGI. For the client optimizer, we use mini-batch SGD throughout. For all tasks, we tune the client learning rate $\eta_l$ and server learning rate $\eta$ by using a large grid search. Full descriptions of the per-task server and client learning rate grids are given in Appendix D.2.

We use the version of FEDADAGRAD, FEDADAM, and FEDYOGI in Algorithm 5. We let $\beta_1 = 0$ for FEDADAGRAD, and we let $\beta_1 = 0.9, \beta_2 = 0.99$ for FEDADAM, and FEDYOGI. For FEDAVG and FEDAVGM, we use Algorithm 4, where CLIENTOPT, SERVEROPT are SGD. For FEDAVGM, the server SGD optimizer uses a momentum parameter of 0.9. For FEDADAGRAD, FEDADAM, and FEDYOGI, we tune the parameter $\tau$ in Algorithm 5.

When tuning parameters, we select the best hyperparameters $(\eta_l, \eta, \tau)$ based on the average training loss over the last 100 communication rounds. Note that at each round, we only see a fraction of the total users (10 for each task except Stack Overflow NWP, which uses 50). Thus, the training loss at a given round is a noisy estimate of the population-level training loss, which is why we averave over a window of communication rounds.

### D.2   HYPERPARAMETER GRIDS

Below, we give the client learning rate ($\eta_l$ in Algorithm 1) and server learning rate ($\eta$ in Algorithm 1) grids used for each task. These grids were chosen based on an initial evaluation over the grids

$$\eta_l \in \{10^{-3}, 10^{-2.5}, 10^{-2}, \ldots, 10^{0.5}\}$$

$$\eta \in \{10^{-3}, 10^{-2.5}, 10^{-2}, \ldots, 10^1\}$$

These grids were then refined for Stack Overflow LR and EMNIST AE in an attempt to ensure that the best client/server learning rate combinations for each optimizer was contained in the interior of the learning rate grids. We generally found that these two tasks required searching larger learning rates than the other two tasks. The final grids were as follows:

**CIFAR-10**:

$$\eta_l \in \{10^{-3}, 10^{-2.5}, \ldots, 10^{0.5}\}$$

$$\eta \in \{10^{-3}, 10^{-2.5}, \ldots, 10^1\}$$

**CIFAR-100**:

$$\eta_l \in \{10^{-3}, 10^{-2.5}, \ldots, 10^{0.5}\}$$

$$\eta \in \{10^{-3}, 10^{-2.5}, \ldots, 10^1\}$$

**EMNIST AE**:
$$\eta_l \in \{10^{-1.5}, 10^{-1}, \ldots, 10^2\}$$
$$\eta \in \{10^{-2}, 10^{-1.5}, \ldots, 10^1\}$$

**EMNIST CR**:
$$\eta_l \in \{10^{-3}, 10^{-2.5}, \ldots, 10^{0.5}\}$$
$$\eta \in \{10^{-3}, 10^{-2.5}, \ldots, 10^1\}$$

**Shakespeare**:
$$\eta_l \in \{10^{-3}, 10^{-2.5}, \ldots, 10^{0.5}\}$$
$$\eta \in \{10^{-3}, 10^{-2.5}, \ldots, 10^1\}$$

**StackOverflow LR**:
$$\eta_l \in \{10^{-1}, 10^{-0.5}, \ldots, 10^3\}$$
$$\eta \in \{10^{-2}, 10^{-1.5}, \ldots, 10^{1.5}\}$$

**StackOverflow NWP**:
$$\eta_l \in \{10^{-3}, 10^{-2.5}, \ldots, 10^{0.5}\}$$
$$\eta \in \{10^{-3}, 10^{-2.5}, \ldots, 10^1\}$$

For all tasks, we tune $\tau$ over the grid:

$$\tau \in \{10^{-5}, \ldots, 10^{-1}\}.$$

### D.3 PER-TASK BATCH SIZES

Given the large number of hyperparameters to tune, and to avoid conflating variables, we fix the batch size at a per-task level. When comparing centralized training to federated training in Section 5, we use the same batch size in both federated and centralized training. A full summary of the batch sizes is given in Table 7.

Table 7: Client batch sizes used for each task.

| TASK | BATCH SIZE |
|------|------------|
| CIFAR-10 | 20 |
| CIFAR-100 | 20 |
| EMNIST AE | 20 |
| EMNIST CR | 20 |
| SHAKESPEARE | 4 |
| STACKOVERFLOW LR | 100 |
| STACKOVERFLOW NWP | 16 |

### D.4 BEST PERFORMING HYPERPARAMETERS

In this section, we present, for each optimizer, the best client and server learning rates and values of $\tau$ found for the tasks discussed in Section 5. Specifically, these are the hyperparameters used in Figure Figure 1 and table Table 1. The validation metrics in Table 1 are obtained using the learning rates in Table 8 and the values of $\tau$ in Table 9. As discussed in Section 4, we choose $\eta, \eta_l$ and $\tau$ to be the parameters that minimizer the average training loss over the last 100 communication rounds.

Table 8: The base-10 logarithm of the client ($\eta_l$) and server ($\eta$) learning rate combinations that achieve the accuracies from Table 1. See Appendix D.2 for a full description of the grids.

|  | FEDADAGRAD | | FEDADAM | | FEDYOGI | | FEDAVGM | | FEDAVG | |
|---|---|---|---|---|---|---|---|---|---|---|
|  | $\eta_l$ | $\eta$ | $\eta_l$ | $\eta$ | $\eta_l$ | $\eta$ | $\eta_l$ | $\eta$ | $\eta_l$ | $\eta$ |
| CIFAR-10 | -³⁄₂ | -1 | -³⁄₂ | -2 | -³⁄₂ | -2 | -³⁄₂ | -½ | -½ | 0 |
| CIFAR-100 | -1 | -1 | -³⁄₂ | 0 | -³⁄₂ | 0 | -³⁄₂ | 0 | -1 | ½ |
| EMNIST AE | ³⁄₂ | -³⁄₂ | 1 | -³⁄₂ | 1 | -³⁄₂ | ½ | 0 | 1 | 0 |
| EMNIST CR | -³⁄₂ | -1 | -³⁄₂ | -⁵⁄₂ | -³⁄₂ | -⁵⁄₂ | -³⁄₂ | -½ | -1 | 0 |
| SHAKESPEARE | 0 | -½ | 0 | -2 | 0 | -2 | 0 | -½ | 0 | 0 |
| STACKOVERFLOW LR | 2 | 1 | 2 | -½ | 2 | -½ | 2 | 0 | 2 | 0 |
| STACKOVERFLOW NWP | -½ | -³⁄₂ | -½ | -2 | -½ | -2 | -½ | 0 | -½ | 0 |

Table 9: The base-10 logarithm of the parameter $\tau$ (as defined in Algorithm 2) that achieve the validation metrics in Table 1.

|  | FEDADAGRAD | FEDADAM | FEDYOGI |
|---|---|---|---|
| CIFAR-10 | -2 | -3 | -3 |
| CIFAR-100 | -2 | -1 | -1 |
| EMNIST AE | -3 | -3 | -3 |
| EMNIST CR | -2 | -4 | -4 |
| SHAKESPEARE | -1 | -3 | -3 |
| STACKOVERFLOW LR | -2 | -5 | -5 |
| STACKOVERFLOW NWP | -4 | -5 | -5 |

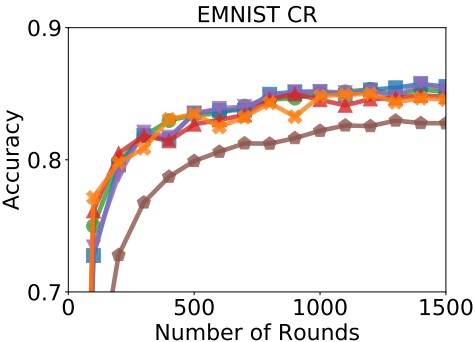

Figure 4: Validation accuracy on EMNIST CR using constant learning rates $\eta$, $\eta_l$, and $\tau$ tuned to achieve the best training performance on the last 100 communication rounds; see Appendix D for hyperparameter grids.

Table 10: Test set performance for Stack Overflow tasks after training: Accuracy for NWP and Recall@5 ($\times 100$) for LR. Performance within within $0.5\%$ of the best result are shown in bold.

| FED... | ADAGRAD | ADAM | YOGI | AVGM | AVG |
|---|---|---|---|---|---|
| STACK OVERFLOW NWP | 24.4 | **25.7** | **25.7** | 24.5 | 20.5 |
| STACK OVERFLOW LR | **66.8** | 65.2 | 66.5 | 46.5 | 40.6 |

## E    ADDITIONAL EXPERIMENTAL RESULTS

### E.1    RESULTS ON EMNIST CR

We plot the validation accuracy of FEDADAGRAD, FEDADAM, FEDYOGI, FEDAVGM, FEDAVG, and SCAFFOLD on EMNIST CR. As in Figure 1, we tune the learning rates $\eta_l, \eta$ and adaptivity $\tau$ by selecting the parameters obtaining the smallest training loss, averaged over the last 100 training rounds. The results are given in Figure 4.

We see that all methods are roughly comparably throughout the duration of training. This reflects the fact that the dataset is quite simple, and most clients have all classes in their local datasets, reducing any heterogeneity among classes. Note that SCAFFOLD performs slightly worse than FEDAVG and all other methods here. As discussed in Section 5, this may be due to the presence of stale client control variates, and the communication-limited regime of our experiments.

### E.2    STACK OVERFLOW TEST SET PERFORMANCE

As discussed in Section 5, in order to compute a measure of performance for the Stack Overflow tasks as training progresses, we use a subsampled version of the test dataset, due to its prohibitively large number of clients and examples. In particular, at each round of training, we sample 10,000 random test samples, and use this as a measure of performance over time. However, once training is completed, we also evaluate on the full test dataset. For the Stack Overflow experiments described in Section 5, the final test accuracy is given in Table 10.

### E.3    LEARNING RATE ROBUSTNESS

In this section, we showcase what combinations of client learning rate $\eta_l$ and server learning rate $\eta$ performed well for each optimizer and task. As in Figure 2, we plot, for each optimizer, task, and pair $(\eta_l, \eta)$, the validation set performance (averaged over the last 100 rounds). As in Section 5, we fix $\tau = 10^{-3}$ throughout. The results, for the CIFAR-10, CIFAR-100, EMNIST AE, EMNIST CR,

Shakespeare, Stack Overflow LR, and Stack Overflow NWP tasks are given in Figures 5, 6, 7, 8, 9, 10, and 11, respectively.

While the general trends depend on the optimizer and task, we see that in many cases, the adaptive methods have rectangular regions that perform well. This implies a kind of robustness to fixing one of $\eta, \eta_l$, and varying the other. On the other hand, FEDAVGM and FEDAVG often have triangular regions that perform well, suggesting that $\eta$ and $\eta_l$ should be tuned simultaneously.

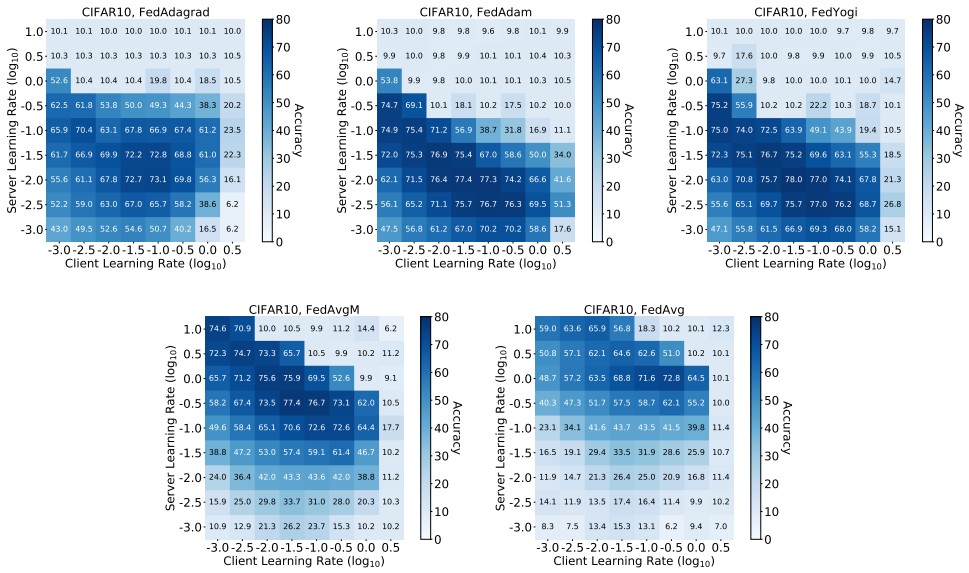

Figure 5: Validation accuracy (averaged over the last 100 rounds) of FEDADAGRAD, FEDADAM, FEDYOGI, FEDAVGM, and FEDAVG for various client/server learning rates combination on the CIFAR-10 task. For FEDADAGRAD, FEDADAM, and FEDYOGI, we set $\tau = 10^{-3}$.

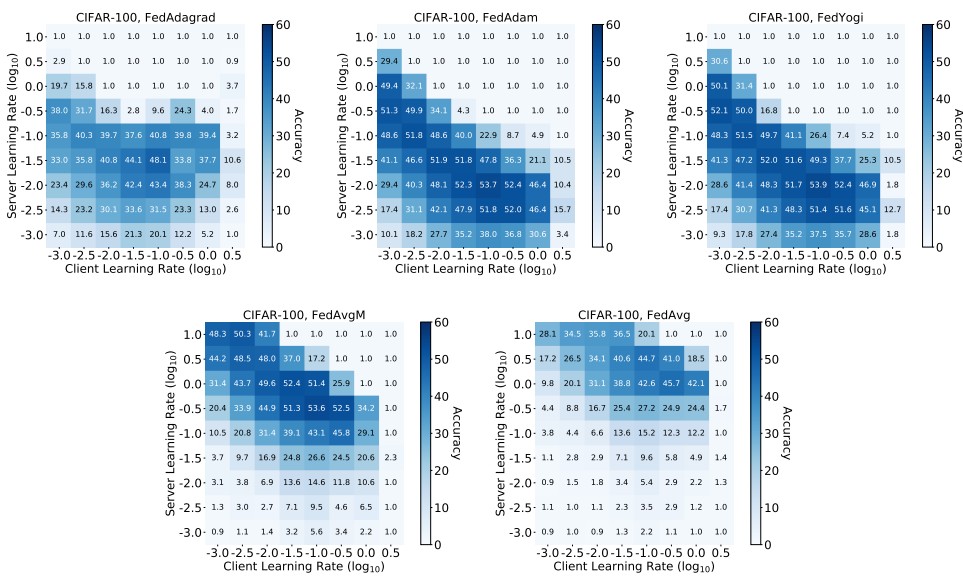

Figure 6: Validation accuracy (averaged over the last 100 rounds) of FEDADAGRAD, FEDADAM, FEDYOGI, FEDAVGM, and FEDAVG for various client/server learning rates combination on the CIFAR-100 task. For FEDADAGRAD, FEDADAM, and FEDYOGI, we set $\tau = 10^{-3}$.

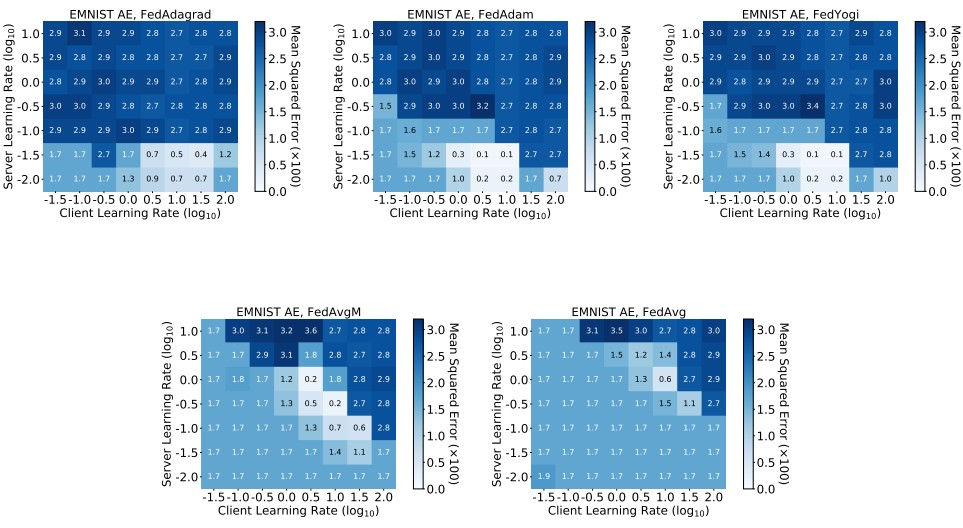

Figure 7: Validation MSE (averaged over the last 100 rounds) of FEDADAGRAD, FEDADAM, FEDYOGI, FEDAVGM, and FEDAVG for various client/server learning rates combination on the EMNIST AE task. For FEDADAGRAD, FEDADAM, and FEDYOGI, we set $\tau = 10^{-3}$.

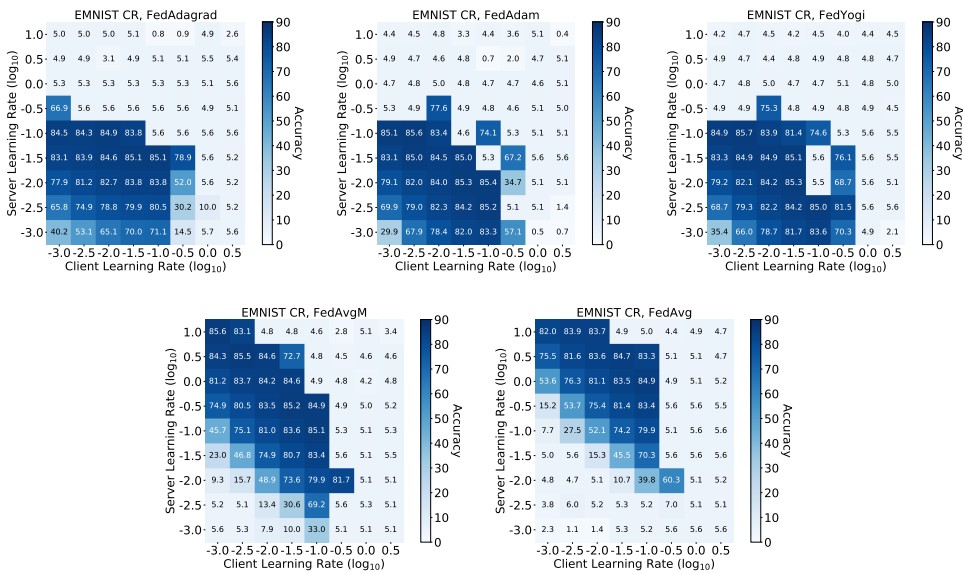

Figure 8: Validation accuracy (averaged over the last 100 rounds) of FEDADAGRAD, FEDADAM, FEDYOGI, FEDAVGM, and FEDAVG for various client/server learning rates combination on the EMNIST CR task. For FEDADAGRAD, FEDADAM, and FEDYOGI, we set $\tau = 10^{-3}$.

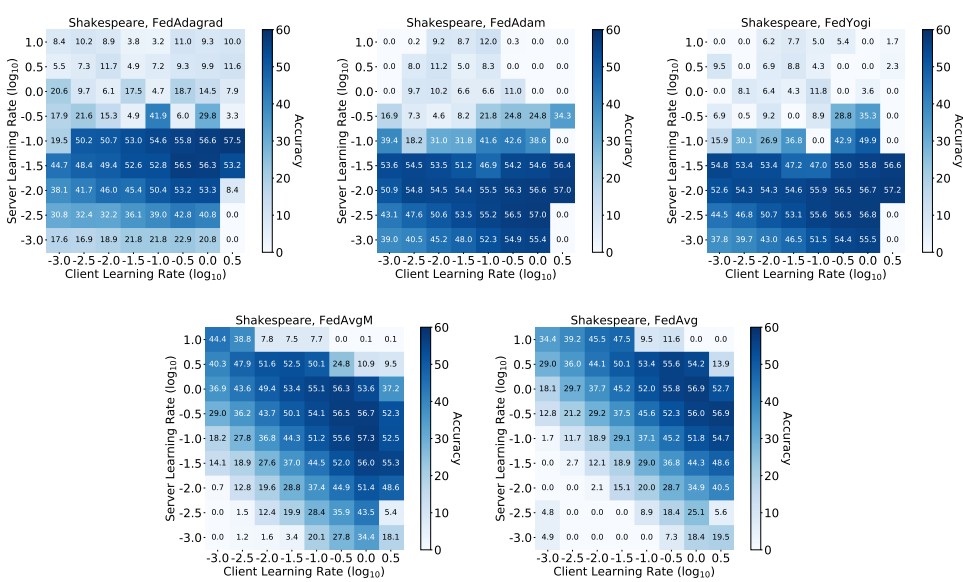

Figure 9: Validation accuracy (averaged over the last 100 rounds) of FEDADAGRAD, FEDADAM, FEDYOGI, FEDAVGM, and FEDAVG for various client/server learning rates combination on the Shakespeare task. For FEDADAGRAD, FEDADAM, and FEDYOGI, we set $\tau = 10^{-3}$.

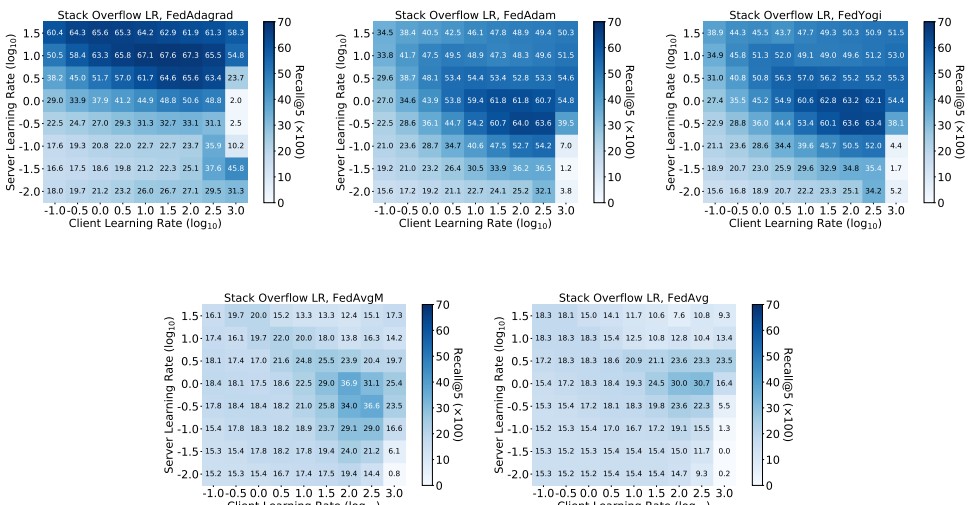

Figure 10: Validation recall@5 (averaged over the last 100 rounds) of FEDADAGRAD, FEDADAM, FEDYOGI, FEDAVGM, and FEDAVG for various client/server learning rates combination on the Stack Overflow LR task. For FEDADAGRAD, FEDADAM, and FEDYOGI, we set $\tau = 10^{-3}$.

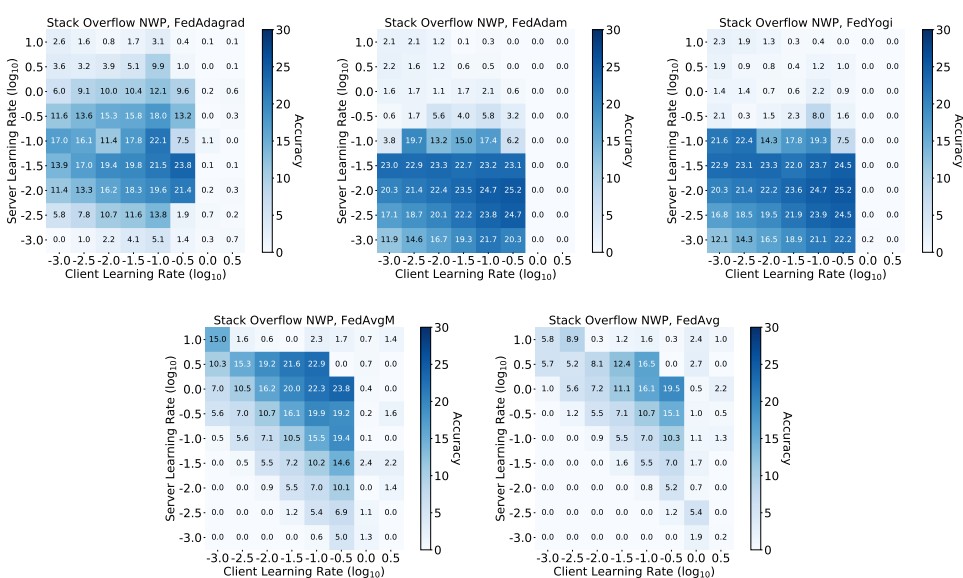

Figure 11: Validation accuracy (averaged over the last 100 rounds) of FEDADAGRAD, FEDADAM, FEDYOGI, FEDAVGM, and FEDAVG for various client/server learning rates combination on the Stack Overflow NWP task. For FEDADAGRAD, FEDADAM, and FEDYOGI, we set $\tau = 10^{-3}$.

### E.4 ON THE RELATION BETWEEN CLIENT AND SERVER LEARNING RATES

In order to better understand the results in Appendix E.3, we plot the relation between optimal choices of client and server learning rates. For each optimizer, task, and client learning rate $\eta_l$, we find the best corresponding server learning rate $\eta$ among the grids listed in Appendix D.2. Throughout, we fix $\tau = 10^{-3}$ for the adaptive methods. We omit any points for which the final validation loss is within 10% of the worst-recorded validation loss over all hyperparameters. Essentially, we omit client learning rates that did not lead to any training of the model. The results are given in Figure 12.

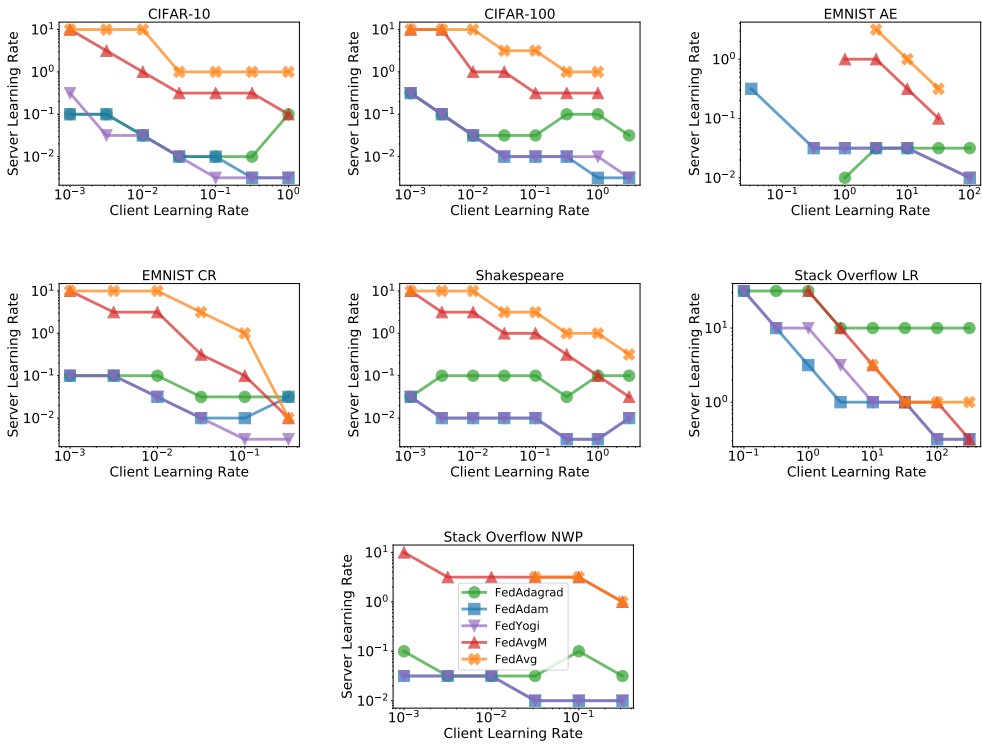

Figure 12: The best server learning rate in our hyperparameter tuning grids for each client learning rate, optimizers, and task. We select the server learning rates based on the average validation performance over the last 100 communication rounds. For FEDADAGRAD, FEDADAM, and FEDYOGI, we fix $\tau = 10^{-3}$. We omit all client learning rates for which all server learning rates did not change the initial validation loss by more than 10%.

In virtually all tasks, we see a clear inverse relationship between client learning rate $\eta_l$ and server learning rate $\eta$ for FEDAVG and FEDAVGM. As discussed in Section 5, this supports the observation that for the non-adaptive methods, $\eta_l$ and $\eta$ must in some sense be tuned simultaneously. On the other hand, for adaptive optimizers on most tasks we see much more stability in the best server learning rate $\eta$ as the client learning rate $\eta_l$ varies. This supports our observation in Section 5 that for adaptive methods, tuning the client learning rate is more important.

Notably, we see a clear exception to this on the Stack Overflow LR task, where there is a definitive inverse relationship between learning rates among all optimizers. The EMNIST AE task also displays somewhat different behavior. While there are still noticeable inverse relationships between learning rates for FEDAVG and FEDAVGM, the range of good client learning rates is relatively small. We emphasize that this task is fundamentally different than the remaining tasks. As noted by Zaheer et al. (2018), the primary obstacle in training bottleneck autoencoders is escaping saddle points, not in converging to critical points. Thus, we expect there to be qualitative differences between EMNIST AE and other tasks, even EMNIST CR (which uses the same dataset).

### E.5 ROBUSTNESS OF THE ADAPTIVITY PARAMETER

As discussed in Section 5, we plot, for each adaptive optimizer and task, the validation accuracy as a function of the adaptivity parameter $\tau$. In particular, for each value of $\tau$ (which we vary over $\{10^{-5}, \dots, 10^{-1}\}$, see Appendix D), we plot the best possible last-100-rounds validation set performance. Specifically, we plot the average validation performance over the last 100 rounds using the best a posteriori values of client and server learning rates. The results are given in Figure 13.

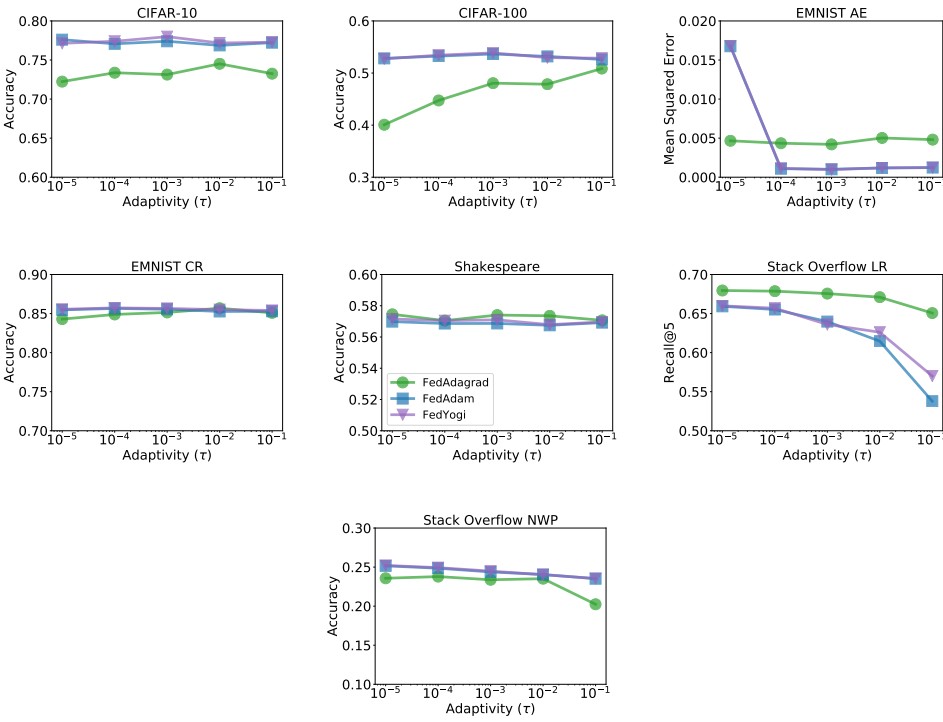

Figure 13: Validation performance of FEDADAGRAD, FEDADAM, and FEDYOGI for varying $\tau$ on various tasks. The learning rates $\eta$ and $\eta_l$ are tuned for each $\tau$ to achieve the best training performance on the last 100 communication rounds.

### E.6 IMPROVING PERFORMANCE WITH LEARNING RATE DECAY

Despite the success of adaptive methods, it is natural to ask if there is still more to be gained. To test this, we trained the EMNIST CR model in a centralized fashion on a shuffled version of the dataset. We trained for 100 epochs and used tuned learning rates for each (centralized) optimizer, achieving an accuracy of 88% (see Table 11, CENTRALIZED row), significantly above the best EMNIST CR results from Table 1. The theoretical results in Section 3 point to a partial explanation, as they only hold when the client learning rate is small or is decayed over time.

To validate this, we ran the same hyperparameter grids on the federated EMNIST CR task, but using a client learning rate schedule. We use a "staircase" exponential decay schedule (EXPDECAY) where the client learning rate $\eta_l$ is decreased by a factor of 0.1 every 500 rounds. This is analogous in some sense to standard staircase learning rate schedules in centralized optimization (Goyal et al., 2017). Table 11 gives the results. EXPDECAY improves the accuracy of all optimizers, and allows most to get close to the best centralized accuracy. While we do not close the gap with centralized optimization entirely, we suspect that further tuning of the amount and frequency of decay may lead to even better accuracies. However, this may also require performing significantly more communication rounds, as the theoretical results in Section 3 are primarily asymptotic. In communication-limited settings, the added benefit of learning rate decay seems to be modest.

Table 11: (Top) Test accuracy (%) of a model trained centrally with various optimizers. (Bottom) Average test accuracy (%) over the last 100 rounds of various federated optimizers on the EMNIST CR task, using constant learning rates or the EXPDECAY schedule for $\eta_l$. Accuracies (for the federated tasks) within $0.5\%$ of the best result are shown in bold.

| | ADAGRAD | ADAM | YOGI | SGDM | SGD |
|---|---|---|---|---|---|
| CENTRALIZED | 88.0 | 87.9 | 88.0 | 87.7 | 87.7 |

| FED... | ADAGRAD | ADAM | YOGI | AVGM | AVG |
|---|---|---|---|---|---|
| CONSTANT $\eta_l$ | 85.1 | 85.6 | 85.5 | 85.2 | 84.9 |
| EXPDECAY | 85.3 | **86.2** | **86.2** | **85.8** | 85.2 |

## F  CREATING A FEDERATED CIFAR-100

**Overview**    We use the Pachinko Allocation Method (PAM) (Li & McCallum, 2006) to create a federated CIFAR-100. PAM is a topic modeling method in which the correlations between individual words in a vocabulary are represented by a rooted directed acyclic graph (DAG) whose leaves are the vocabulary words. The interior nodes are topics with Dirichlet distributions over their child nodes. To generate a document, we sample multinomial distributions from each interior node's Dirichlet distribution. To sample a word from the document, we begin at the root, and draw a child node its multinomial distribution, and continue doing this until we reach a leaf node.

To partition CIFAR-100 across clients, we use the label structure of CIFAR-100. Each image in the dataset has a *fine label* (often referred to as its label) which is a member of a *coarse label*. For example, the fine label "seal" is a member of the coarse label "aquatic mammals". There are 20 coarse labels in CIFAR-100, each with 5 fine labels. We represent this structure as a DAG $G$, with a root whose children are the coarse labels. Each coarse label is an interior node whose child nodes are its fine labels. The root node has a symmetric Dirichlet distribution with parameter $\alpha$ over the coarse labels, and each coarse label has a symmetric Dirichlet distribution with parameter $\beta$.

We associate each client to a document. That is, we draw a multinomial distribution from the Dirichlet prior at the root $(\mathrm{Dir}(\alpha))$ and a multinomial distribution from the Dirichlet prior at each coarse label $(\mathrm{Dir}(\beta))$. To create the client dataset, we draw leaf nodes from this DAG using Pachinko allocation, randomly sample an example with the given fine label, and assign it to the client's dataset. We do this 100 times for each of 500 distinct training clients.

While more complex than LDA, this approach creates more realistic heterogeneity among client datasets by creating correlations between label frequencies for fine labels within the same coarse label set. Intuitively, if a client's dataset has many images of dolphins, they are likely to also have pictures of whales. By using a small $\alpha$ at the root, client datasets become more likely to focus on a few coarse labels. By using a larger $\beta$ for the coarse-to-fine label distributions, clients are more likely to have multiple fine labels from the same coarse label.

One important note: Once we sample a fine label, we randomly select an element with that label *without replacement*. This ensures no two clients have overlapping examples. In more detail, suppose we have sample a fine label $y$ with coarse label $c$ for client $m$, and there is only one remaining such example $(x, c, y)$. We assign $(x, c, y)$ to client $m$'s dataset, and remove the leaf node $y$ from the DAG $G$. We also remove $y$ from the multinomial distribution $\theta_c$ that client $m$ has associated to coarse label $c$, which we refer to as *renormalization* with respect to $y$ (Algorithm 8). If $i$ has no remaining children after pruning node $j$, we also remove node $i$ from $G$ and re-normalize the root multinomial $\theta_r$ with respect to $c$. For all subsequent clients, we draw multinomials from this pruned $G$ according to symmetric Dirichlet distributions with the same parameters as before, but with one fewer category.

**Notation and method**    Let $\mathcal{C}$ denote the set of coarse labels and $\mathcal{Y}$ the set of fine labels, and let $\mathcal{S}$ denote the CIFAR-100 dataset. This consists of examples $(x, c, y)$ where $x$ is an image vector, $c \in \mathcal{C}$ is a coarse label set, and $y \in \mathcal{Y}$ is a fine label with $y \in c$. For $c \in \mathcal{C}, y \in \mathcal{Y}$, we let $\mathcal{S}_c$ and $\mathcal{S}_y$ denote the set of examples in $\mathcal{S}$ with coarse label $c$ and fine label $y$. For $v \in G$ we let $|G[v]|$ denote the set of children of $v$ in $G$. For $\gamma \in \mathbb{R}$, let $\mathrm{Dir}(\gamma, k)$ denote the symmetric Dirichlet distribution with $k$ categories.

---

**Algorithm 7** Creating a federated CIFAR-100

---

Input: $N, M \in \mathbb{Z}_{>0}, \alpha, \beta \in \mathbb{R}_{\geq 0}$
**for** $m = 1, \cdots, M$ **do**
    Sample $\theta_r \sim \text{Dir}(\alpha, |G[r]|)$
    **for** $c \in \mathcal{C} \cap G[r]$ **do**
        Sample $\theta_c \sim \text{Dir}(\beta, |G[c]|)$
    $D_m = \emptyset$
    **for** $n = 1, \cdots N$ **do**
        Sample $c \sim \text{Multinomial}(\theta_r)$
        Sample $y \sim \text{Multinomial}(\theta_c)$
        Select $(x, c, y) \in \mathcal{S}$ uniformly at random
        $D_m = D_m \cup \{(x, c, y)\}$
        $\mathcal{S} = \mathcal{S} \backslash \{(x, c, y)\}$
        **if** $\mathcal{S}_y = \emptyset$ **then**
            $G = G \backslash \{y\}$
            $\theta_c = \text{RENORMALIZE}(\theta_c, y)$
            **if** $\mathcal{S}_c = \emptyset$ **then**
                $G = G \backslash \{c\}$
                $\theta_r = \text{RENORMALIZE}(\theta_r, c)$

---

**Algorithm 8** RENORMALIZE

---

Initialization: $\theta = (p_1, \ldots, p_K), i \in [K]$
$a = \sum_{k \neq i} p_k$
**for** $k \in [K], k \neq i$ **do**
    $p'_k = p_k / a$
Return $\theta' = (p'_1, \cdots p'_{i-1}, p'_{i+1}, \cdots, p'_K)$

---

Let $M$ denote the number of clients, $N$ the number of examples per client, and $D_m$ the dataset for client $m \in \{1, \cdots, M\}$. A full description of our method is given in Algorithm 7. For our experiments, we use $N = 100, M = 500, \alpha = 0.1, \beta = 10$. In Figure 14, we plot the distribution of unique labels among the 500 training clients. Each client has only a fraction of the overall labels in the distribution. Moreover, there is variance in the number of unique labels, with most clients having between 20 and 30, and some having over 40. Some client datasets have very few unique labels. While this is primarily an artifact of performing without replacement sampling, this helps increase the heterogeneity of the dataset in a way that can reflect practical concerns, as in many settings, clients may only have a few types of labels in their dataset.

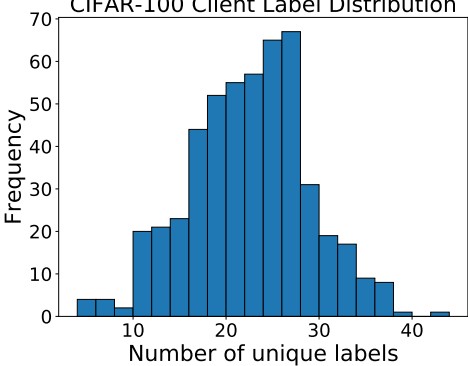

Figure 14: The distribution of the number of unique labels among training client datasets in our federated CIFAR-100 dataset.

