# OpenReview forum: "Adaptive Federated Optimization"
_ICLR.cc/2021/Conference — ICLR 2021 Poster_

### Official Review · AnonReviewer2 · 2020-10-21
**Review of "Adaptive Federated Optimization"**

**Rating:** 6
**Confidence:** 4

**Review:**

Summary: This paper presents an adaptive federated optimization framework that induces three different adaptive federated learning algorithms, which are proposed to address the issues of client drift due to data heterogeneity and lack of adaptivity. The authors presented thorough literature survey on the federated learning and formulated the meta-algorithm, FedOpt, introducing both server and client optimizers. Different from the popular FedAvg, the server optimizer in this work is an adaptive protocol which is originated from the client drift. The authors analyzed mathematically the convergence rates of the proposed framework and showed extensive experimental results on different benchmark datasets to validate the efficacy of the developed algorithms.

Overall, this paper is well organized and technically sound. It is also easy to follow. The overall proof ideas are also correct, though I didn’t derive step by step. However, the authors need to pay attention to the following points to improve the current draft.

1. Server optimizer can be confusing. The server optimizer developed in this study generalizes to a more adaptive one. But it is not really an optimization in the server, instead just another form of local model averaging in order to obtain the global model. I completely understand the motivation for this is to improve the performance beyond FedAvg, which may not necessarily perform well with non-IID data. However, without using zeroth order or first order information of objectives, to me, the so-called server optimizer is just a nonlinear local model averaging.
2. The server learning rate is bit counterintuitive. In Corollary 1 and 2, when defining the learning rate for the server, it looks like $\eta$ can be quite large. Of course, given the fact that the server optimizer is not really an optimization, that could be understandable. Additionally, the authors empirically investigated the relationship between the server and client learning rates in the appendix. But to me, it is naturally a parameter that plays a similar role as learning rate, not exactly the learning rate.
3. The communication efficiency needs more discussion. In Section 3, the authors presented one discussion on the communication efficiency, particularly quantifying $K$. I wonder how the authors obtained that. Also, it would be great to see more quantitative results on the communication efficiency.
4. Client heterogeneity is only qualitative in the paper. Based on the empirical implementation, I couldn’t find out how the authors simulated the non-IID data distributions for different clients for each benchmark task. This needs more detail. Also, the authors mentioned that the proposed framework can work effectively for moderate and naturally arising heterogeneity. But under how much heterogeneity does the algorithms work well? For example, for CIFAR 10, if having 10 clients, each client has only one class without any overlapping, is FedOpt still effective?
5. The experimental results look promising, but not showing significant outperforming capability. From Table 1, we can still observe that for some tasks, the AvgM is still favorably comparable. Also, when to select which method among FedAdam, FedAdaGrad, and FedYOGI is unclear. The authors need to give more discussion.
6. Based on the Corollary 1 and 2, it looks like when $T$ is sufficiently large, the FedOpt can achieve linear speed up, correct? However, the authors failed to give the specific lower bound for $T$, even if they only mentioned the dominating term that induced a sublinear rate.

***********************************
After carefully considering the rebuttal from the authors, I think I am more positive about the paper so I raised my score. The rebuttal clarifies most of my confusion about the paper, though more improvement can be done.

---

> ### Author Response · Authors · 2020-11-18
> **Regarding "server optimization", communication-efficiency, and client heterogeneity**
>
> We thank the reviewer for their time and comments, and are glad that the reviewer enjoyed the organization and writing of the paper. Some discussion of the issues raised by the reviewer:
>
> (1) One of the core ideas of our work is that we can use any first-order optimization method as SERVEROPT in FedOpt. As mentioned in Section 2, our paper shows that the local updates from the clients can be used as “pseudo-gradients” with a wide range of optimization methods (such as Adam, Adagrad, Yogi) used as SERVEROPT. This justifies the terminology "server optimizer". Our analysis crucially relies on properties of first-order methods and these pseudo-gradients (see bounds $T_1$, $T_2$ in the proof of Thm 1). We fully acknowledge that this was not made clear before, and that the use of the term "optimizer” is somewhat overloaded; we have added discussion to address this (see the end of Section 2).
>
> (2) Our previous point highlights the justification of using server optimizer terminology. In such a scenario, the parameter $\eta$ is akin to a server learning rate, though as the reviewer rightly pointed out, this parameter is similar but not identical to a learning rate.
>
> (3) The quantification of K discussed in Section 3 can be obtained from Corollary 1 & 2. First, recall that $\sigma^2 = \sigma_l^2 + 6K \sigma_g^2$. The terms $\sigma^2/GKT$ and $\sigma^2/\sqrt{K}T^{3/2}$ in Corollary 1 & 2 are dominant when K is large ($K \gg T$ unless $\sigma_l/\sigma_g$ is large). This imposes a constraint on K in the analysis. As noted in the paper, $\sigma_l/\sigma_g \to \infty$ corresponds to the i.i.d case, in which case K can be very large.
>
> We agree that further analysis on how to select K is an interesting study. However, choosing K is an open question even for FedAvg. We expect that a solution to this requires new tools and techniques that bridge the gap between asymptotic and practical convergence. However, our experiments already showcase some communication-efficiency benefits. Since our methods have the same communication cost as FedAvg, they obtain better performance for fixed communication budgets.
>
> (4) Our empirical analysis primarily concerns how our methods work under realistic data heterogeneity settings. We discuss the distribution of the datasets in great detail (see Appendix C). As we note there, for Shakespeare, Stack Overflow and EMNIST, the partitioning of data among clients is intrinsic (eg. each client in Stack Overflow corresponds to a user of the website). These datasets (with the client partitioning) are publicly available from the LEAF and TensorFlow Federated repositories. For the CIFAR datasets, we describe explicitly how we partition data among clients (see Appendix C.1).
>
> Regarding “under how much heterogeneity does the algorithms work well”: This is difficult to answer, partly because there is no standard measure of heterogeneity. Our focus is on a more immediately practical question: "Do adaptive methods perform well on real-world examples of heterogeneity?" As to whether they would work when we have 10 clients, each with a single (non-overlapping) label, we believe this is not a useful benchmark. Each client could learn a perfect model by training on its own data. We believe that using FL to learn a single global model is only useful under moderate heterogeneity. In cases with extreme and pathological heterogeneity, personalization approaches or purely local training should perform much better (for detailed discussion, see Section 3.3.4 of Kairouz et al., 2019).
>
> (5) While there are tasks where FedAvgM performs comparably to adaptive methods, there are tasks (eg. Stack Overflow LR) where it performs much worse. As we show in Figure 2, it is also easier to find good client/server learning rate combinations for adaptive methods than FedAvgM. As we discuss in the “Dense-gradient tasks” paragraph of Section 5, FedAdam and FedYogi do at least as well, if not significantly better, than all other methods throughout. As a result we recommend the two (which perform comparably) in all instances. We have updated our paper to make this recommendation clear (see the end of “Dense-gradient tasks”).
>
> (6) As we discuss in point (i) after Corollary 2, the dominant term in the convergence rate is on the order of $1/\sqrt{mKT}$. Thus, we do not achieve a linear speed up. This is the best known rate in general non-convex settings (e.g. see Karimireddy et al., 2019). Regarding lower bounds: For AdaGrad (Corollary 1), we need $T \ge max(64, 20 (L/G)^{2/3} (Km)^{1/3})$, for Adam (Corollary 2), we need $T \ge max(36, 36 (L/G)^{1/2} (Km)^{1/4})$.
>
> -----
> Peter Kairouz, H Brendan McMahan, et al. Advances and open problems in federated learning. arXiv preprint arXiv:1912.04977, 2019.
>
> Sai Praneeth Karimireddy, Satyen Kale, Mehryar Mohri, Sashank J Reddi, Sebastian U Stich, and Ananda Theertha Suresh. SCAFFOLD: Stochastic controlled averaging for on-device federated learning. arXiv preprint arXiv:1910.06378, 2019.

---

### Official Review · AnonReviewer3 · 2020-10-28
**Characterizes the convergence of adaptive methods for federated learning; the necessity of using adaptive methods remains open**

**Rating:** 6
**Confidence:** 4

**Review:**

This paper studies the convergence of well-known adaptive methods, ADAM, ADAGRAD, and YOGI, for the federated learning problem. In particular, while the nodes (clients) still use SGD for their local computations (same as Fed-Avg), the server uses one of the three adaptive methods mentioned above to update the model. The authors have provided convergence rates (based on gradient norms) for all three methods for the nonconvex settings. Moreover, various experiments have been conducted to compare the strengths of these methods against classic FedAvg. Overall, this paper is well-written, and the problem statement and the goal of the paper are clear. Moreover, the authors have shown the success of using adaptive methods in numerical settings.

My main questions are regarding the theoretical analysis of the paper. First, the analysis of adaptive methods in the nonconvex setting has been studied in the literature (as mentioned by the authors as well in the related work). Given the bounded gradient assumption imposed in this paper, I am not sure what would be the main challenge in extending those analyses to the federated setting. I encourage the authors to highlight possible novelties in their analysis compared to prior works on adaptive methods for nonconvex objective functions.


Second, it is not clear to me how this paper's theoretical results would imply that adaptive methods enjoy better convergence rates compared to regular Fed-Avg. In particular, all the algorithms achieve the rate of $O(1/\sqrt{T})$, and do not differ from each other in that aspect. To better highlight my point, for instance, let's consider the convex setting. There, Duchi et al. show that while AdaGrad has $O(1/\sqrt{T})$ convergence rate in the worst case (same as SGD), it can achieve better rates in certain scenarios, such as gradients being sparse over coordinates, etc. Hence, this way, the theoretical result shows the advantage of using adaptive methods over the standard SGD method. From my understanding, this paper does not provide that kind of results, and so one important question remains unanswered: Why switching to adaptive methods after all? I would appreciate it if the authors provide further discussions on this matter.

-----

Post rebuttal comment: I appreciate the authors' responses, especially on highlighting the theoretical challenges. I have raised my score accordingly.

---

> ### Author Response · Authors · 2020-11-18
> **Regarding the challenges of analyzing adaptive methods for federated learning**
>
> We thank the reviewer for their review and constructive criticism. The main concerns of the review are regarding the theoretical analysis of the paper, which we explain below.
>
> (1) Main challenge in our theoretical analysis: Theoretical analysis for federated learning settings is very different from the centralized settings. The key components of federated learning analysis are: (i) data heterogeneity, (ii) multiple local updates at the client and (iii) communication complexity. As discussed in the paper, the updates from the client optimizer can be considered are not actual gradients but rather pseudo-gradients (see Section 2). The convergence analysis for adaptive methods using pseudo gradient is challenging because they can have both high bias and variance due to local client updates and are tricky to bound. This is especially challenging in the presence of data heterogeneity ($\sigma_g$ is one proxy for data heterogeneity). Moreover, obtaining a good tradeoff between the number of local steps and communication complexity is also another challenging component of the analysis. For instance, see discussion in Section 3 for a trade-off between T and the number of local steps K. We have added this discussion to Appendix A.1 in order to clarify the theoretical contributions of our work.
>
> (2) Regarding better convergence rates: To our knowledge, obtaining better convergence rates for adaptive methods when compared to SGD in general nonconvex settings of our interest is open even in the centralized settings. In convex settings, Adagrad can perform better than SGD in presence of sparse gradients (as the reviewer rightly points out). However, in nonconvex settings of our interest, we are not aware of any works that show a better convergence rate even for Adagrad. To our knowledge, as pointed in our main contributions, ours is the first work to develop an adaptive method for FL and it shows convergence in *federated non-convex* settings. Our analysis shows that the upper bound on convergence rate of adaptive methods is no worse than SGD. Currently (for both centralized & federated non-convex settings), switching to adaptive methods is primarily motivated by empirical performance of adaptive methods. Our compelling empirical evidence shows a strong case for using adaptive federated optimization. In all our experiments, we found FedAdam & FedYogi to perform at least as well as FedAvg and significantly better in many cases. We also found adaptive methods to be robust to learning rates (see Figure 2). Furthermore, there is strong evidence that adaptive methods are necessary in some cases (e.g. heavy tail noise scenario in Zhang et al., 2019 in which case SGD do not even converge). We leave this scenario as a future work.
>
> -----
>
> Jingzhao Zhang, Sai Praneeth Karimireddy, Andreas Veit, Seungyeon Kim, Sashank J. Reddi, Sanjiv Kumar, and Suvrit Sra. Why ADAM beats SGD for attention models. arXiv preprint arxiv:1912.03194, 2019.

---

### Official Review · AnonReviewer1 · 2020-10-29
**The relationship between the proposed FedAdagrad/FedYogi/FedAdam and FedSGD**

**Rating:** 6
**Confidence:** 5

**Review:**

This paper proposes several federated variants of adaptive stochastic gradient methods.   Moreover,  the convergence rates of the proposed algorithms are also provided.  Based on the current submission,  The reviewer has several concerns:

1. The adaptive learning rates in FedAdagrad/FedYogi/FedAdam are all related to parameter $\tau$.  According to Theorem 1-2, Corollary 1-2,   $\tau$ needs to set as a large constant $G/L$.   When  $\tau$ is sufficiently large, FedAdagrad/FedYogi/FedAdam is actually reduced to FedSGD. Can the authors give several comments on this parameter?

2. The second question is about the generalization ability of federated adaptive stochastic gradient methods.   It has been demonstrated in [1] that adaptive SGD generalizes poorly compared with SGD.  A natural question is that "  does this dilemma still exists in federated stochastic gradient methods? "

[1] Wilson AC, Roelofs R, Stern M, Srebro N, Recht B. The marginal value of adaptive gradient methods in machine learning. In Advances in neural information processing systems 2017 (pp. 4148-4158).

---

> ### Author Response · Authors · 2020-11-18
> **Regarding the tau parameter and generalization**
>
> We thank the reviewer for their review and interesting questions. The main concern seems to be the role of the tau parameter and generalization of the adaptive methods, which we discuss below.
>
> (1) On the $\tau$ parameter: $\tau$ controls the amount of adaptivity in adaptive optimization methods. As the reviewer rightly points out, as $\tau$ gets larger, the adaptive methods become less adaptive and behave more like SGD. In our analysis $\tau$ is chosen as $G/L$ to obtain a *sufficient* condition for convergence.  However, one can obtain convergence for smaller values and obtain different constants in the convergence rates. Also, $G/L$ is not necessarily large e.g. when $G$ (the per-coordinate gradient bound) is small compared to $L, G/L$ can indeed be small.
>
> A more subtle point is whether $\tau$ should be lower bounded by some positive real number (rather than setting $\tau = 0$ or sending $\tau \to 0$ as $T \to \infty$). A similar question also arises in the centralized (i.e. non-federated learning settings). For methods like Adam, Zaheer et al., 2018 show through compelling empirical analysis that moderately large $\tau$ (i.e., limited adaptivity) generally yields better optimization and generalization performance (see e.g. Section 4 of Zaheer et al., 2018). We found that to be true even in federated learning settings. Thus, our theoretical analysis reflects settings which also yield good empirical performance. We revised the paper to reflect this discussion (see the second paragraph in the “Ease of tuning” portion of Section 5).
>
> (2) Generalization of adaptive methods: This question is interesting but beyond the scope of the paper. Unfortunately, the question of generalization of adaptive methods even in centralized settings is still open. Wilson et al., 2017 show a pathological case where adaptive methods generalize worse than SGD. However, there are also scenarios that necessitate the use of adaptive methods since SGD does not even converge (e.g. see the heavy tail noise scenario in Zhang et al., 2019 which is typically encountered in NLP tasks). In our experiments, we see that the adaptive methods do not suffer from any generalization issues (when compared to SGD). A more rigorous theoretical characterization of generalization of these methods is interesting, but beyond the scope of this work.
>
> -----
> Wilson AC, Roelofs R, Stern M, Srebro N, Recht B. The marginal value of adaptive gradient methods in machine learning. In Advances in neural information processing systems 2017 (pp. 4148-4158).
>
> Manzil Zaheer, Sashank Reddi, Devendra Sachan, Satyen Kale, and Sanjiv Kumar. Adaptive methods for nonconvex optimization. In Advances in Neural Information Processing Systems, pp. 9815– 9825, 2018.
>
> Jingzhao Zhang, Sai Praneeth Karimireddy, Andreas Veit, Seungyeon Kim, Sashank J. Reddi, Sanjiv Kumar, and Suvrit Sra. Why ADAM beats SGD for attention models. arXiv preprint arxiv:1912.03194, 2019.

---

### Official Review · AnonReviewer4 · 2020-10-29
**A general framework for cross-device FL, focus on adaptive server optimizers, interpretable convergence bounds, comparison to adaptive client optimizer and experiments can be strengthened.**

**Rating:** 7
**Confidence:** 4

**Review:**

This paper extends the server model averaging step in FedAvg to a more general adaptive optimization step on the global model, specifically, by writing the model averaging as a gradient descent step using a pseudo gradient. Three variants of this scheme (FedOpt) are presented, based on three adaptive optimizers, including AdaGrad, ADAM, and YOGI. While there exist works applying the server-side momentum method, the paper argues that the proposed framework is more general since any adaptive optimizer can be applied to extend FedAvg. Under three assumptions (Lipschitz gradient, bounded gradients, bounded variances), the paper provides a local convergence analysis with nonconvex objectives. The achieved bounds match the best-known convergence rate for FL under reasonable conditions (i.e., T is sufficiently large compared to K). They also provide guidance for how to decay the local learning rate to avoid client drift, and show that increasing local steps can help to reduce the communication rounds (under certain conditions). They also reflect the dependency of these bounds on client heterogeneity measured by global variance and indicates how to mitigate the problem in FedOpt. Experimental comparison to FedAvg, FedAvgM, and SCAFFOLD on seven benchmark FL tasks show that the proposed three adaptive FL methods are better or comparable on early-stage convergence and final validation-set performance. A study about tuning the local and global learning rate is also presented.

Pros:

(1) The paper is well written and easier to understand for most parts. The authors moved the proof and experiment details to the appendix, which largely improves the readability.

(2) The proposed FedOpt, as an extension of FedAvg, can incorporate adaptive optimizers for the server-side model update, and thus provide a general framework that can potentially cover a family of FL algorithms, which can be an important contribution to the FL community.

(3) The interpretation of the bounds is also informative: showing several expected or preferred trade-offs in FL and sheds light on how to reduce the communication costs and the negative effects of client heterogeneity.

(4) The experiments show the compelling performance of the proposed methods, especially on Stack Overflow LR and EMNIST AE, on which the improvements over the other two baselines are remarkable.

Cons and suggestions:

(1) One main novelty and focus of this paper is the server-side adaptive optimization, and the purpose is to combat client drift and reduce the variance of stochastic gradient, especially in the cross-device FL setting. However, is it the case that improving client-side optimization can be more effective in resolving these two issues? FedOpt covers the client-side optimization in its framework but did not expand it on possible options, while other works like FedProx focus more on it. It would be helpful to provide a discussion and comparison of the two strategies and their advantages/disadvantages on the two issues.

(2) The theoretical bounds show the trade-off between T and K, the client and server learning rates, and the client heterogeneity. In experiments, only different options of client and server learning rates have been studied. It would be more sound to show the other trade-offs indicated by the bounds.

(3) In experiments, FedAvgM is a very competitive baseline since on most tasks, its performance is very close to the proposed methods, except on SO LR and EMNIST AE. Considering its similarity in methodology with FedOpt, it reduces the novelty of the paper.

(4) There are only three FL baselines that have been compared in the experiments. Is there a specific reason for not including other FL methods into the comparison? In addition, it is interesting to compare with (i) FedOpt with both adaptive client optimizer (or FedProx) and adaptive server optimizer and (2) FedOpt with only adaptive client optimizer.

---

> ### Author Response · Authors · 2020-11-18
> **On comparisons to client adaptivity and FedAvgM**
>
> We thank the reviewer for their time and insightful comments, and are glad that the reviewer enjoyed the organization and writing of the paper. Some discussion of the issues raised by the reviewer:
>
> (1) We believe client-side adaptivity is an important open question, but we wanted to ensure our methods did not increase communication costs relative to FedAvg. Client adaptive methods (such as Local AdaAlter, see Xie et al., 2019) often require extra communication to average client optimizer states. How to do adaptive client optimization without increasing communication costs is an important question, but one we aim to tackle in the future.
>
> (2) We agree that future studies of how to set the parameter K are important. However, we believe this is a subtle, challenging question. Our results in Section 3 are asymptotically the same as convergence rates for FedAvg. However, our experiments show that we can actually do better than FedAvg in many settings. Thus, the interplay between T, K, and the learning rates will likely look much different in practical scenarios than in our theory. Studying the impact of K will require new techniques and tools, which we leave to future work. We also note that how to choose K optimally for FedAvg is still an open question.
>
> (3) Adaptive methods are significantly better in multiple tasks (especially those on the Stack Overflow dataset). Moreover, we show in Figure 2 (Appendix E.3 in full detail) that any similarity in accuracy belies a difference in how easy the methods are to tune. Adaptive methods are frequently much easier to tune. Thus, we believe the novelty is not just in designing methods that achieve better accuracy, but in trying to make FL methods work better with less tuning.
>
> (4) As discussed in (1), we constrained ourselves to methods with the same communication cost as FedAvg. It is currently an open problem how to use adaptive client optimization while still maintaining this same communication cost. We consciously did not use any adaptive client methods in our experiments in order to maintain the same communication costs as FedAvg.
>
> We did not compare to FedProx because our evaluations were focused on settings where each client does a single epoch of local training, and there are no systems-failures. As shown by Li et al., 2018 (see Figure 9, Row 1) in such settings FedProx and FedAvg perform nearly identically.
>
>
> -----
> Cong Xie, Oluwasanmi Koyejo, Indranil Gupta, Haibin Lin. Local AdaAlter: Communication-Efficient Stochastic Gradient Descent with Adaptive Learning Rates. arXiv preprint arxiv:1911.09030, 2019.
>
> Tian Li, Anit Kumar Sahu, Manzil Zaheer, Maziar Sanjabi, Ameet Talwalkar, Virginia Smith. ◊. arXiv preprint arxiv:1812.06127, 2018.

---

### Comment · ~Eric_James1 · 2021-05-02
**Some errors in the proof**

Thanks for the great work.

It looks there are some errors in Lemma 4. You forgot to include the second term $\eta_{l}^{2} K^{2}\left\|\frac{\nabla f\left(x_{t}\right)}{\tau}\right\|^{2}$ of Eq.(11) in Lemma 4.

I guess the conclusions in theorem 1 and 2 may not hold.

---

### Decision · Program_Chairs · 2021-01-07
**Final Decision**

**Decision:**

Accept (Poster)

**Comment:**

The paper proposes adaptive optimization algorithms for federated learning that are federated versions of existing adaptive algorithms such as Adam, Adagrad, and Yogi. The paper establishes convergence guarantees for the proposed algorithms and performs an extensive experimental evaluation. Following the discussion, the reviewers were positive about the paper and felt that the author responses addressed their concerns. I recommend accept.